# Chain of Thoughtlessness?
# An Analysis of CoT in Planning

**Kaya Stechly**[*]
SCAI, Arizona State University
kstechl@asu.edu

**Karthik Valmeekam**[*]
SCAI, Arizona State University
kvalmeek@asu.edu

**Subbarao Kambhampati**
SCAI, Arizona State University
rao@asu.edu

## Abstract

Large language model (LLM) performance on reasoning problems typically does not generalize out of distribution. Previous work has claimed that this can be mitigated with chain of thought prompting–a method of demonstrating solution procedures–with the intuition that it is possible to in-context teach an LLM an algorithm for solving the problem. This paper presents a case study of chain of thought on problems from Blocksworld, a classical planning domain, and examines the performance of two state-of-the-art LLMs across two axes: generality of examples given in prompt, and complexity of problems queried with each prompt. While our problems are very simple, we only find meaningful performance improvements from chain of thought prompts when those prompts are exceedingly specific to their problem class, and that those improvements quickly deteriorate as the size $n$ of the query-specified stack grows past the size of stacks shown in the examples. We also create scalable variants of three domains commonly studied in previous CoT papers and demonstrate the existence of similar failure modes. Our results hint that, contrary to previous claims in the literature, CoT's performance improvements do *not* stem from the model learning general algorithmic procedures via demonstrations but depend on carefully engineering highly problem specific prompts. This spotlights drawbacks of chain of thought, especially the sharp tradeoff between possible performance gains and the amount of human labor necessary to generate examples with correct reasoning traces.[†]

## 1 Introduction

While originally designed for text completion, Large Language Models (LLMs) have shown promise on a diverse set of unrelated tasks. While initial anecdotal results were unexpectedly impressive [8], followup systematic studies showed that–outside of limited, non-generalizable classes of problems– these models generally perform poorly on basic, multi-hop reasoning tasks [17] ranging from arithmetic [35] and logic puzzles [14] to constraint satisfaction [42, 2] and classical planning [47].

At the same time, the subfield of prompt engineering [36] has grown rapidly, promising improvements in performance without retraining. A core tenet of this subfield is that LLMs are capable of powerful

---

[*]equal contribution

[†]Resources and source code for planning experiments can be found at `https://github.com/karthikv792/cot-planning` and for other domains at `https://github.com/kstechly/cot-scheduling`

in-context learning [12, 56], that is, capable of intelligently using additional context provided in a prompt to correctly respond to queries that would otherwise be answered incorrectly. Generally, this requires operationalizing *algorithmic/procedural advice*, and, in principle, learning such procedures includes being able to effectively apply them beyond syntactically similar instances.

The foundational method for inducing in-context learning is the chain of thought approach, which has been claimed to "unlock the reasoning abilities of LLMs" [50]. To create a chain of thought (CoT) prompt, a user annotates similar problems with intermediate reasoning steps and prepends them to the standard prompt. These annotations are meant as demonstrations, intended to teach a procedure applicable to both the examples and the new query. When prompted like this, the LLM is expected to output a similar series of reasoning steps prior to the new answer. Numerous studies have claimed that this procedure significantly enhances LLM performance in complex reasoning tasks [49, 54, 39, 56, 52, 43]. However, in general it is unclear how "similar" the examples need to be to the problem, how broadly any given chain of thought prompt will apply, and–most importantly–how much human effort is necessary to craft prompts specific to each problem subclasses. Followup work has claimed that merely adding magic phrases ("let's think step by step") to every prompt is sufficient for some improvement [26]. While in some domains, this technique has proven to be even more brittle than manual CoT, it has achieved the same performance increases in others, hinting that improvements observed with CoT may not indicate as much about LLMs' general in-context learning abilities as previously thought.

We are interested in the tradeoff between possible performance gains from chain of thought prompt engineering and the amount of human labor necessary to generate examples with useful reasoning traces. Ideally, a properly constructed prompt should teach the LLM how to robustly generalize a basic algorithmic procedure in order to increase performance on a large class of problems, thereby converting a modest amount of human teaching effort into a significant capability boost. Unfortunately, this only seems to be possible to a very limited extent [14].

In the current work, we examine the limits of chain of thought in solving classical planning problems. Test domains commonly used in previous chain of thought studies (e.g. GSM8K [10], CommonSense QA [44]) present two significant issues: (a) they lack a systematic method to scale instances, which is essential for evaluating whether LLMs can extend provided procedures to larger instances of the same type, and (b) due to their static nature, are more likely to be well-represented on the web[51], increasing the chance that they were part of LLM training data, a factor which could obscure the true reasoning capabilities of LLMs. Planning is a well-studied kind of sequential decision-making which tasks an agent with devising a plan that takes a given initial state to a pre-specified goal state. New, diverse, and unique problem instances are easy to generate, but potentially hard to solve.

We focus on Blocksworld, a simple commonsense domain widely recognized and utilized in International Planning Competitions [23], where a set of blocks in an initial configuration must be rearranged step-by-step into a goal configuration. For a subset of our results, we simplify even further, and only consider problem instances where every block starts on the table and the goal is a single stack of blocks. These instances require very minimal reasoning: one need only figure out which block is on the bottom, and then stack the remaining blocks in the sequence directly defined in the goal. For $3 \leq n \leq 20$, we generate a variety of instances where the goal requires a specific $n$ height stack, while providing examples of how to solve 2 and 3 height instances.

We consider different chain of thought prompts, where each is more specific–and provides more problem-specific knowledge–than the last: a zero-shot variant, a general progression proof, a suboptimal algorithm specific to Blocksworld, a table-to-stack specific simplification of that algorithm, and a lexicographic version of the simplification. The most general could be applied to any problem, while the least is specific to an easier version of the stacking problem. The three human-crafted prompts all teach algorithms which could, in principle, solve any of the instances they are tested on. We test on three state of the art models: GPT-4 [3], Claude-3-Opus, [5] and GPT-4-Turbo.

Our results reconfirm that LLMs are generally incapable of solving simple planning problems [47], and demonstrate that chain of thought approaches only improve performance when the hand-annotated examples and the query are sufficiently similar to the current query. As goal stack size increases, accuracy drops drastically, regardless of the specificity of the chain of thought prompt. As generality of the prompt increases, performance on even the smallest goal stacks also decreases, and often falls short of standard prompting. Even state of the art extensions of CoT (like self-consistency [49]), show similar or sometimes even worse performance. Overall, this case study calls into question

assumptions about the generalizable effectiveness of chain of thought, and suggests that LLMs do not learn new, general algorithms in context, but instead rely on some form of pattern matching to achieve prompt-design-specific performance increases. This in turn increases the burden on humans giving advice.

To better compare to previous work, we construct scalable versions of three previously studied synthetic problems–Coin Flip, Last Letter Concatenation, and multi-step arithmetic [49, 50, 26, 48]– and replicate reported chain of thought prompts. While these domains do not have a corresponding notion of prompt granularity, they do cover a range of difficulties. When testing on GPT-4-Turbo, We see a similar lack of generalization on these problem sets as we saw in Blocksworld.

In the rest of this paper, we first review related work, then describe the chain of thought approaches we have developed in the context of planning, analyze the overall effectiveness of chain of thought prompting on Blocksworld problems, and extend our results to three synthetic tasks well-represented in the CoT literature.

## 2   Related Work

Modifying text prompts to elicit intermediate problem-solving steps from LLMs originally took the form of scratchpads [33]. [50] proposed a similar prompt style in natural language, dubbing this approach chain of thought (CoT), and claiming that–with some human hand-annotation of examples–this not only boosts performance without retraining, but "allows reasoning abilities to emerge naturally". They argued that by merely interspersing intermediate reasoning steps in natural language into examples, they were inducing the LLM to "learn via a few examples", motivating this idea with anthropomorphizations ("Consider one's own thought process when solving a complicated reasoning task such as a multi-step math word problem"). [26] argued that some of the performance of CoT could be retained without providing any examples, and instead just appending the magic phrase "let's think step by step" to the end of a prompt. This has been called zero-shot CoT.

However, CoT has long been known to be imperfect and incomplete. Previous work has investigated improving the consistency of CoT through self-consistency [49], multi-agent debate [13], least-to-most prompting [55], deductive verification [28], and other approaches. Unfortunately, many of these involve prompting the LLM multiple times for a single problem, which can balloon the cost of inference. Other work has examined the possibility of reducing or removing the need for human annotation of examples by using LLMs to generate their own examples automatically [54, 9]. To avoid well-known issues with the brittleness of LLM self-verification and self-teaching [42, 22, 20, 19, 24], we restrict this paper's scope to manually written chains of thought.

Previous papers have analyzed CoT from multiple perspectives [15, 37], finding that there is only a loose relationship between the presented chain and the final answer [6], and that the correctness of provided annotations has little effect on resultant performance [38]. LLM-produced chains of thought are also known to be unfaithful to the underlying reasoning process [29, 25, 11]. In particular, the way the examples are presented can bias a model into giving some answer (e.g. if all the example answers are A, the model will be more likely to output A), but its CoT will not reflect this [45].

Motivated by claims that CoT prompts allow models to learn in context how to reason–that is, to learn how to execute human-specified algorithms–we focus on CoT prompting's out-of-domain generalization. [14] previously showcased a lack of generalization in multiplication, puzzles, and a number sequence problem, even when the model was fine-tuned on CoT examples. However, they only examined one set of prompts, did not experiment with levels of prompt specificity, and were much more interested in local failures of compositionality arising from cumulating error. More broadly, previous work has examined generalization limits of LLMs in arithmetic tasks [35], formula simplification [34], and theorem proving [4].

While early accounts claimed LLMs, despite not being trained for it, were capable of reasoning and planning [8], later work showcased serious brittleness across these domains [47]. [50] claims that "standard prompting only provides a lower bound on the capabilities of large language models", with proper prompting allowing reasoning to "emerge naturally." Recent work seems to maintain this optimism [7]. In this paper, we examine the effectiveness of CoT in the context of classical planning problems, which have well-defined and algorithmically checkable ground truths, can be generated with arbitrary size and difficulty, and are unlikely to be in the training data. If CoT induces more than just pattern matching, and can in fact teach LLMs to perform generalizable, compositional reasoning,

then we should expect that to be reflected in robust and maintainable improvements on a simple commonsense benchmark set like Blocksworld, and we should expect these results to hold for scaled variants of the very benchmarks tested in [50] and later CoT work.

# 3    Background

Classical planning problems task a planner with finding a sequence of actions that, when executed, will take an agent from a pre-specified initial state to a desired goal state. STRIPS planning is a discrete, deterministic formalism that encompasses this class. Problems are represented using the Planning Domain and Definition Language (PDDL) [30] and have long featured in various planning challenges and competitions. Our main experiments are all on the Blocksworld PDDL domain.

A PDDL specification consists of three components. The *domain* doesn't change between problems and consists of a set of predicates–whose truth values describe the state of the world–and a set of actions–defined by their preconditions and effects–that the agent is allowed to take. The *initial state* is a list of predicates that are true at the outset of the specific problem (an example predicate: "Block A is on the table"). The *goal* is a boolean expression of predicates (a goal: "Block A is on Block B.").

A *plan* is a sequence of actions. The solution to a PDDL problem is a plan in which the preconditions of every action are satisfied at execution time, and which arrives at a goal-satisfying final state. To verify a plan, follow the actions in order and check that these two desiderata are achieved. In this work, we convert natural language responses into PDDL [46] and evaluate them with VAL [21].

# 4    Chain of Thought Setups for Planning

We examine the influence of prompt selection on LLM performance within subsets of the Blocksworld domain. A formally specified problem instance can be translated into many possible prompts. The most basic of these is input/output (I/O) prompting: the problem is translated directly from PDDL into natural language and provided to the LLM [47]. While this directly tests the LLM's ability to solve the problem, it is not always the most effective strategy for maximizing performance.

Drawing on metaphors of human learning, recent literature has claimed that LLMs are capable of *in-context learning*. The basic idea is that–by first presenting the model with examples of similar problems–it is possible to cause an LLM to acquire relevant new skills within the current context window. $n$-shot prompts operationalize this by prepending a number of relevant examples. *Chain of thought* [50] approaches take this further, presenting human-crafted "thoughts" which the LLM is intended to imitate in its response. Practitioners argue that, intuitively, these augmented examples teach the LLM *how* to solve problems in the given set.

Figure 1: Target Distributions of Problems. This figure shows the levels of expected generality for each prompt.

However, this method relies on human labor [53] to provide task-specific knowledge and an (at least rough) algorithmic or procedural approach to the problem. The more general the provided knowledge is, the more problems it can be applied to, and the less human prompt-crafting it requires. On the other hand, the more granular and specific it is, the more performance can be expected.

In our experiments, we consider subsets of Blocksworld problems. We follow a prompt structure similar to that described in [47], [1] but include "thoughts" in our $n$-shot prompts. These thoughts are written to follow an algorithmic procedure for solving the example problem.

Not every procedure is applicable to every problem. From the point of view of a human hand-crafting a chain of thought prompt, there is intuitively an *expected* target distribution on which the

---

[1]Prompt and response examples for each approach can be found in the Appendix.

demonstrated algorithm generally works. For instance, a prompt designer detailing how to stack C on top of B on top of A will expect that a model that learns this procedure will also be capable of stacking B on top of A on top of C, but may not expect it to know how to first properly dismantle an existing tower of blocks to access a necessary block. However, this distribution often differs from the *effective* target distribution–that is, the actual set of problems on which the prompt gives robust improvements in performance. We explicitly describe the gap between these two distributions.

**Zero-Shot Chain of Thought (Universal):** This is the most general approach, and involves merely appending "let's think step by step" to the end of the prompt[26].

**Progression Proof (Specific to PDDL):** Versions of this CoT could, in principle, be prepended to any PDDL problem prompt, as the generation of annotated examples is easy to automate without knowledge of the specific PDDL domain. [47] This prompt includes (1) a meta-prompt explaining plan correctness and (2) an example where each action is annotated with the state prior to the action, the reason why the action is applicable in that state, and the resulting state after the action is applied. Examples start from an arbitrary block configuration and construct a single stack of blocks from it.

**Blocksworld Universal Algorithm (Specific to the Domain):** In Blocksworld, it is possible to reach any goal state from any initial state by simply unstacking all the blocks, placing them on the table, and then reassembling them into the required stacks. Resulting plans are not only executable and goal-reaching, but will never exceed twice the length of the optimal plan for any given instance [40]. This prompt demonstrates an annotated version of this approach, explaining and performing both the deconstruction and reconstruction steps of the algorithm. The same examples are used as in the previous prompt. The expected target distribution encompasses all Blocksworld problems.

**Stacking Prompt (Specific to a Narrow Problem Class):** Every example is a table-to-stack problem: every block starts on the table, and the goal is to create a *single* specific stack of blocks. This specificity simplifies the problem greatly, and allows near-direct pattern matching between the examples and the LLM's output; however, it is infeasible to specify prompts with this level of detail for every problem class. The expected target distribution is table-to-stack Blocksworld problem, as they are the only problems that can be solved by the described algorithm.

**Lexicographic Stacking (Specific to Particular Syntactic Sequences):** We simplify the problem further by focusing on a particular syntactic form of the goal. This prompt is very similar to the stacking prompt, but is specific to a subset of the target distribution: the goal state is always a lexicographic prefix (e.g., A, AB, ABC, etc.).

## 5 Blocksworld Results

We perform two parallel studies. The first tests each chain of thought prompt on its intended problem distribution, as explained in the previous section. Then, we focus on a specific subclass of Blocksworld problems and test every prompt on just that subclass. Together, we expect these two studies to give us a good picture of how effective LLMs are in applying advice beyond the specific instances.

### 5.1 Testing on Intended Problem Distributions

We evaluate the performance of GPT-4 and Claude-3-Opus on Blocksworld problems with both standard 2-shot prompts and chain of thought prompts of varying granularity. Each prompt is tested on its intended problem class, as discussed in the previous section.

Chain of thought does not meaningfully enhance performance except on the narrowest problem distributions. While providing this chain of thought advice becomes significantly harder as the level of specificity increases, it is necessary, as the LLM succeeds only when the problem is reduced to a level where basic pattern matching suffices: at each stage, stack the next letter on top; if that letter does not exist on the table, then stop.

A key advantage of planning domains is that they provide the ability to easily and systematically generate larger test sets, including arbitrarily more challenging instances. The difficulty of a Blocksworld instance scales with the number of blocks involved, allowing us to clearly assess the out-of-domain generalization achievable with and without chain of thought. As shown in Figure 2, chain of thought

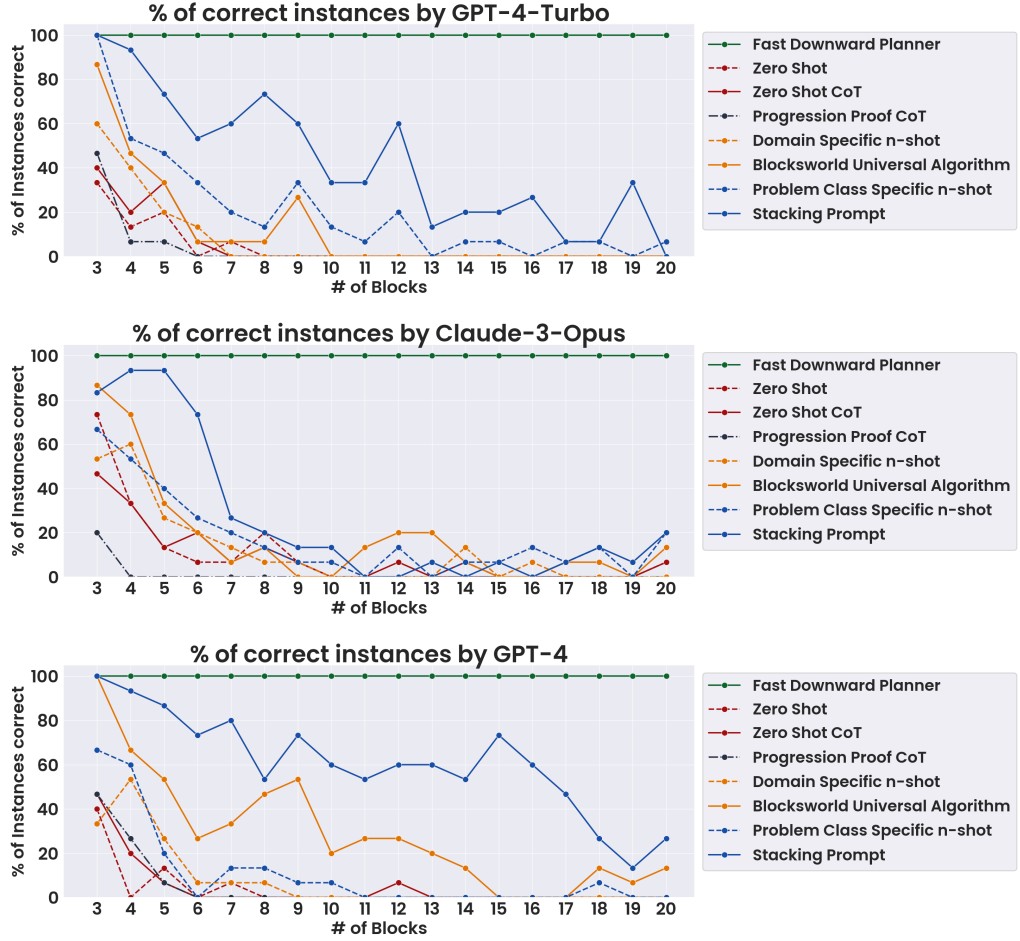

Figure 2: Accuracy of GPT-4-Turbo, Claude-3-Opus and GPT-4 across chain of thought prompting methods in their intended problem distributions with increasing number of blocks.

does not generalize beyond a handful of blocks. Note that sound planning systems (such as Fast Downward) have a 100% accuracy on all problems tested.

## 5.2 Testing only on Table-to-Stack

As mentioned before, a table-to-stack problem is any problem in the intended target distribution of the stacking prompt. The initial state has every block on the table, with a goal of arranging all the blocks into a single, pre-specified stack. While a simple problem, GPT-4's zero-shot performance over 261 instances is 3.8%. With the stacking CoT prompt, performance improves to 59.3%. Is this a result of the model learning in-context how to reason correctly over this type of problem? If so, we might expect it to perform the same when presented with a more general CoT prompt that demonstrates the same procedure, but is applicable to a greater variety of problems.

To check this, we evaluate performance of our prompts on table-to-stack problems with prompts of varying granularity: standard I/O prompting, general $n$-shot (drawn from arbitrary Blocksworld problems), goal-specific $n$-shot (drawn from table-to-stack problems), and three levels of CoT specificity. Table 1 shows the results: only the most specific and least applicable prompt retains anywhere near this performance improvement. Figure A.1.1 in the appendix further illustrates that none of the prompts provide robust stack-height generalizability. We also tested self-consistency[49] on these prompts, but found that performance dropped. Details can be found in Appendix A.2.

| Prompt | GPT-4-Turbo | Claude-3-Opus | GPT-4 |
|---|---|---|---|
| zero-shot | 19.1% | 9.96% | 3.83% |
| zero-shot CoT | 21% | 10.34% | 4.98% |
| Domain-Specific $n$-shot | 13.7% | 16.4% | 6.13% |
| Progression Proof CoT | 15.3% | 4.59% | 6.89% |
| Domain-Specific $n$-shot | 13.7% | 16.4% | 6.13% |
| Blocksworld Universal Algorithm | 37.1% | 37.1% | 51.3% |
| Problem Class Specific $n$-shot | 18% | 15.7% | 8.81% |
| Stacking Prompt | 40.6% | 24.5% | 59.3% |

Table 1: Accuracy across CoT and example granularities over 261 instances in **table-to-stack** Blocksworld.

If chain of thought is meant to replicate human thinking or learning, it should generalize beyond the most direct pattern matches and allow for more robust reasoning across similar problems. However, our results only show a modest improvement in performance on some domains, with specific enough prompting strategies, which quickly deteriorates when the problems shown become slightly larger.

## 6 Extension to Scalable Synthetic Benchmarks

Previous work on CoT mainly constrained its evaluations to static test sets ranging from commonsense domains (Sports Understanding [41], StrategyQA [18], CommonSenseQA [44]), few-hop math word problems (AsDiv [31], GSM8k [10], MAWPS [27]), to a number of basic "symbolic reasoning" tasks (CoinFlip [26], LastLetterConcatenation [26], Shuffled Objects [41]). [26, 50, 55, 6]. Many of these benchmarks are difficult to scale, but a number of them can be modified to allow for the generation of arbitrary new instances which nevertheless have clear ground truths. We examine CoinFlip, LastLetterConcatenation, and a synthetic proxy for multi-step arithmetical reasoning. Exact prompt details can be found in the appendices A.7, A.8, and A.9. When possible we used the manual CoT prompts found in [50] and the zero-shot CoT prompt described in [26]. Number of examples ranges from 0 to 3 for both CoT and direct prompts. Results for all three domains are in Table 2 and Figure 3.

**CoinFlip:** Parity tests have a long history in machine learning[32]. CoinFlip is a natural language version of this task introduced in [50] to showcase the performance of CoT, though that paper only studies up to four flip problems. An example prompt is "A coin is heads up. Craig flips the coin. Alice does not flip the coin. Is the coin still heads up?". The correct answer is "no". Note that chance performance on this domain is 50%, as there are only two possible answers. Our extension to the domain is detailed in A.3

**LastLetterConcatenation:** Also introduced in [50], the LastLetterConcatenation task is a simple text processing task that asks for the concatenation of the last letters of a series of words. An example prompt is "Take the last letters of each word in 'Craig Alice' and concatenate them." for which the correct answer is "ge". The set of possible answers on this task is much larger than in CoinFlip, but previous work has claimed significant performance increases on this kind of task with CoT. Modeling something similar to our Blocksworld granularity experiments, we create two other test sets, using the same underlying words in the same distribution, but which differ in what they ask the model to do. LastVowelConcatenation requires using only the last vowels of words. FoomLetterConcatenation requires using the first letter of the first word, the second letter of the second word, and so forth. If the $n$th word does not have an $n$th letter, the problem specifies that a 0 should be concatenated to the string instead.

**Multi-step Arithmetic on Single-Digit Numbers:** CoT is often tested on math word problems. However, many of these test sets only include problems which require very small numbers of reasoning steps. GSM8k was designed partly so that its problems would "require more steps to solve", but its problems only range 2 to 8 steps[10], and, in fact, previous analyses have found that only 10% of those problems require more than five steps–the majority is 2, 3, or 4. [16]

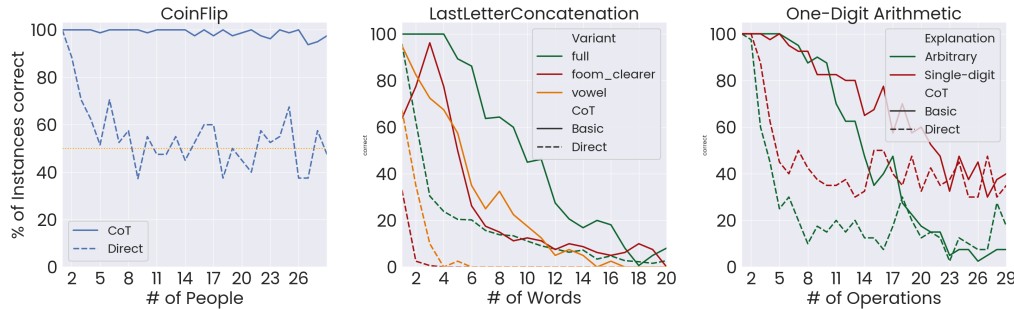

Figure 3: Accuracy of GPT-4-Turbo with chain of thought prompting across variations of our synthetic datasets. "Direct" means direct prompting without any CoT.

| Prompt | CF | LLC | LVC | FLC | Arithmetic | AE |
|--------|------|------|------|------|------------|------|
| Zero-Shot | 56.38% | 10.00% | 5.75% | 1.81% | 24.13% | 45.60% |
| Zero-Shot CoT | 95.71% | 52.54% | N/A | N/A | 56.12% | 42.76% |
| Manual CoT | 98.89% | 51.06% | 27.00% | 26.00% | 50.43% | 69.31% |
| Incorrect Cot | 96.76% | 48.15% | N/A | N/A | N/A | N/A |

Table 2: Accuracy across CoT types and problem variations over all instances in our synthetic datasets. CF is CoinFlip, LLC is LastLetterConcatenation, LVC is LastVowelConcatenation, FLC is FoomLetterConcatenation, Arithmetic is baseline single-digit Arithmetic, AE is the same problems but with the explanation provided that all intermediate answers are single digit.

To sidestep this issue, we construct a synthetic dataset that involves linearly simplifying parenthesized expressions that consist of repeated applications of the four basic arithmetical operations on one digit numbers. An example prompt is "Simplify the following expression into a single number: 3 / (9 - (5 + (1))).", where the correct answer is 1. We filter our problems so that no operation ever results in a number that isn't in the range 1 to 9.[2] This can be seen as a deeply simplified variant of the arithmetical expression simplification dataset presented in [34] where no modular arithmetic, negative numbers, or non-linear nesting is required. However, we extend our maximum number of required reasoning steps much further and we construct prompts which are more specific and spell out every single step explicitly. More details on the dataset can be found in A.5.

## 6.1 Results

**Length Generalization**   The only synthetic domain that shows any hints of generalization is CoinFlip. Using [50]'s prompt, performance is perfect for 1 through 4 step problems, starts to show the occasional mistake after, and only dips below 90% at 31-step problems (as shown in Figure 3). However, the problems in this domain are very simple. Parallel to the lexicographic stacking case of Blocksworld, it does not require much reasoning beyond counting up to an average of half a given problem's step count.

LastLetterConcatenation and multi-step arithmetic show behavior almost identical to our main experiments. While sufficiently specific CoT prompts do increase performance on small instances, this performance increase quickly degrades as the number of steps necessary increases. Notably, the string-based nature of the LastLetterConcatenation problem does allow us to examine what exact improvement CoT is inducing. We examine the data with different metrics and find that the only properties that do generalize with CoT are syntactic. In particular, while overall accuracy plummets back to that of direct prompting, CoT consistently improves the Levenshtein distance to the correct answer and ensures that the final response string contains exactly the right letters, just not in the

---

[2]We exclude 0, since any number multiplied by zero is zero, and this would quickly lead to zero representing around 50% of correct answers for larger numbers of reasoning steps.

right order or number. We take this as further evidence that CoT, rather than teaching algorithms or procedures, modifies the syntactic style of the LLM's output, and that this pattern matching is what leads to observed increases in performance on smaller instances.

**Prompt Granularity and Problem Variation**    Because of the simplicity of these problems, prompt granularity is much harder to examine than in Blocksworld. There isn't enough variation in possible problems. However, across the three types of letter concatenation and two types of arithmetic expression simplification that we test, we see very similar patterns as before: CoT's performance improvements are maintained much longer in easier cases, and take longer to collapse back to direct performance. There still seems to be a "sweet spot" where the problem is just barely hard enough that CoT makes a difference, but not so hard that this difference doesn't matter.

**Examining Intermediate Reasoning**    The construction of our synthetic arithmetic task gives some hints as to what part of CoT may be failing. [14] argues that compositional reasoning fails because LLMs perform linearized subgraph matching and act as noisy estimators of intermediate functions (see e.g. proposition 4.2 in [14]) and that performance collapses follow from the fact that repeated application of any error-prone function estimator leads to exponentially accumulating error.

In our problem, it is possible to exhaustively check whether this is the case. There are exactly 118 possible 1-digit binary arithmetic problems which result in a 1-digit number. We tested GPT-4-Turbo, GPT-4, GPT-3.5-Turbo, Llama3-70b, and Llama3-8b on this dataset at various temperatures and every single model scored 100%. However, despite perfect performance on application of the required intermediate function, CoT still does not lead to robust generalization to arbitrary length problems. Therefore, at least on this problem set, the issue isn't due to accumulating error. The problem must be with the LLM's inability to learn the correct algorithm from contextual demonstrations, rather than with its inability to execute that algorithm.

Overall, we see that our results on planning are not a fluke. These three synthetic domains showcase similar generalization failures, but these failures only become clear when the problems tested on require sufficiently many reasoning steps or when the minor modifications of the domain are studied. This illustrates the need for testing on benchmarks which can generate arbitrary new instances of increasing difficulty. Without such testing, conclusions drawn from static test sets of limited size are unlikely to be robust. We implore the community at large to adopt more rigorous evaluation mechanisms, especially when making claims about the poorly-understood yet much-hyped algorithmic reasoning abilities of black box models.

## 7   Conclusion

In this paper, we conducted a systematic evaluation of the effectiveness of chain of thought in large language models on a specific classical planning problem. Our case study indicates that, contrary to previous claims in the literature, providing examples of procedural reasoning does not induce the general ability to apply that procedure to novel instances in current state-of-the-art large language models. In fact, the performance improvements seen when prompting LLMs in this manner quickly vanish when queries differ in generality from the examples, despite the fact that the same algorithmic procedure applies to the larger or more general instance.Very specific prompts are more likely to work, but they can require significantly more human labor to craft. Our results indicate that chain of thought prompts may only work consistently within a problem class if the problem class is narrow enough and the examples given are specific to that class. Both of these facts show that chain of thought approaches provide less generalization than previous claims seem to indicate, and hint that basic pattern matching rather than in context learning of general algorithmic procedures may better explain the improvements seen from chain of thought.

## 8   Acknowledgements

This research is supported in part by ONR grant N0001423-1-2409, and gifts from Qualcomm, J.P. Morgan and Amazon.

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

# A  Appendix

## Contents

### A.1  Broader Impacts

Chain of Thought (CoT) has become one of the most widely adopted ideas for improving planning and reasoning abilities of LLMs. Almost every system routinely, and uncritically, uses some prompting strategy attributed to CoT. On the flip side, whenever LLMs are shown to have limitations in any sphere, practitioners tend to question those studies by attributing it to unskilled use of CoT methodology. Our study, based on both in planning and other more standard tasks, calls into question the prevalent belief that LLMs are capable of operationalizing and generalizing the CoT advice effectively. It instead suggests that CoT is effective only when the LLM can do straightforward pattern matching between the example and the problem. We believe that the lessons of this study will be helpful in mitigating the applications of LLMs to tasks requring planning and reasoning with false confidence.

### A.2  Self Consistency on Table to Stack problems

We evaluated self consistency [49], a state-of-the-art extension of CoT, on table to stack problems. We sampled 5 different reasoning paths (with temperature 0.7) and chose the most frequent plan breaking ties randomly. As our results show (in Table 3 and Figure A.2.1), self-consistency does not lead to a generalization breakthrough, and in fact is generally *worse* than the original results in Table 1. This is likely because the solution space for planning problems is much larger than that studied in previous (often multiple choice) benchmarks. In fact, most queries led to five unique responses, forcing us to choose the final answer from them at random.

### A.3  Further details on modifications to the CoinFlip domain

Given a list of names, generating new instances is just a matter of filling in a template. We source our list of names from the U.S. Social Security Administration [1], and only keep names with at least 50 occurrences. We scale our problems with the number of names (potentially repeated) mentioned in the prompt. The main test set consists of 1120 instances, with 40 instances per number of names and 28 different numbers of names, ranging from 1 to 28. We also tested [50]'s prompt on an extended

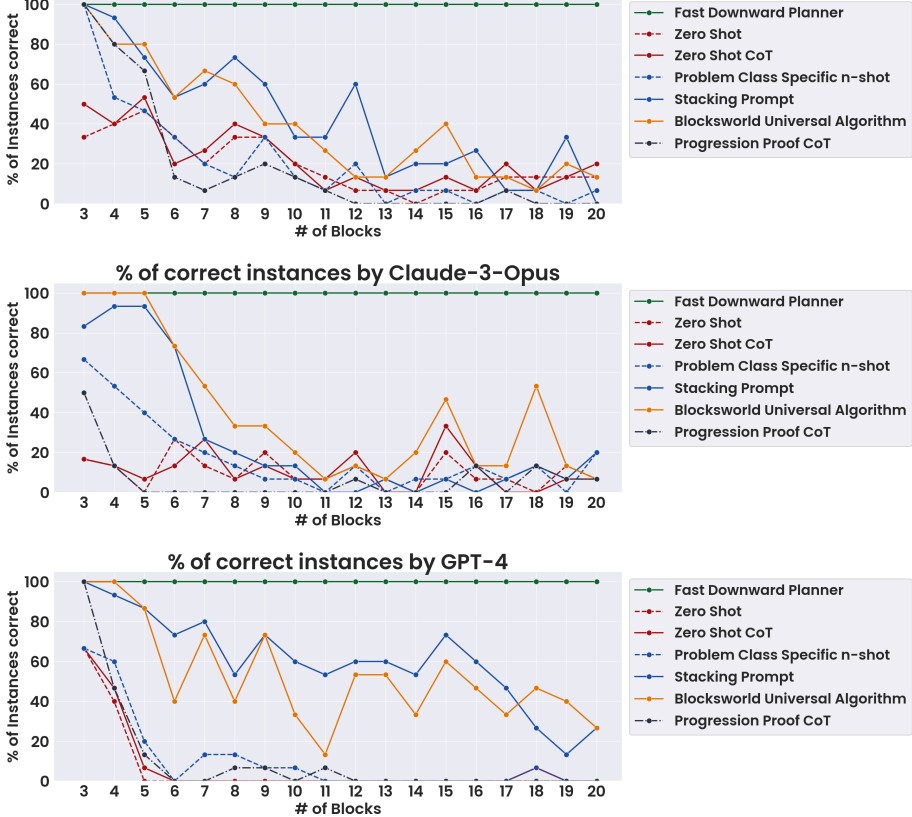

Figure A.1.1: (Table-to-stack) Accuracy of GPT-4-Turbo, Claude-3-Opus and GPT-4 across chain of thought prompting methods with increasing number of blocks.

| Prompt | GPT-4-Turbo | Claude-3-Opus |
|---|---|---|
| zero-shot | 18.3% | 9.19% |
| zero-shot CoT | 18.3% | 11.8% |
| Problem Class Specific $n$-shot | 15.7% | 14.9% |
| Stacking Prompt | 24.5% | 26.4% |

Table 3: Accuracy of Self-consistency over 261 instances in **table-to-stack** Blocksworld.

set of 2960 instances, with 40 instances per number of names, but only stopping at 75 names, finding that performance did begin to decrease more significantly past 30 names.

## A.4  Further details on modifications to the LastLetterConcatenation domain

We use the same database as in CoinFlip to generate words. [1] To scale instances, we simply increase the number of words whose last letters must be concatenated. Our problems range from 1 to 20 words, with 40 instances per word, giving a total of 800 problems.

## A.5  Further details on the multi-step Arithmetic dataset

The number of reasoning steps in this domain corresponds directly to the number of operations that need to be performed to simplify a given expression. Our test set consists of 1160 total problems, spread 1 to 29 operations, with 40 instances per number of operations. Again mirroring

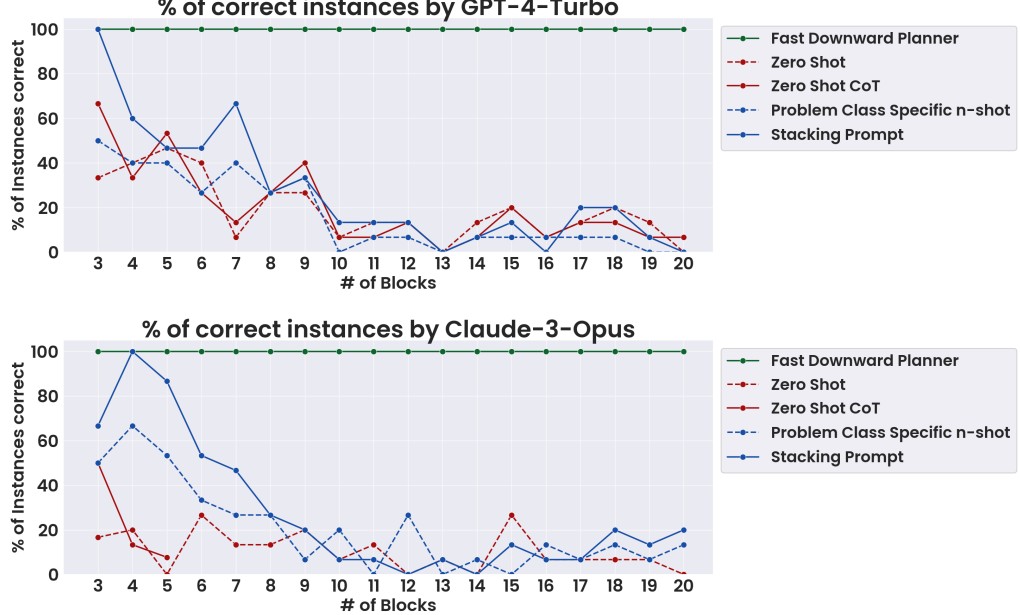

Figure A.2.1: (Table-to-stack) Accuracy of GPT-4-Turbo and Claude-3-Opus across chain of thought prompting methods with self consistency.

our Blocksworld test cases, we experiment with two variants: prompting as if these were any expression simplification problems, and prompting with the explicit additional information that every intermediate step must be a single digit number.

## A.6 Planning Prompts and Responses by GPT-4

### A.6.1 Domain Information

> **Domain Information**
>
> ```
> I am playing with a set of blocks where I need to arrange the blocks into stacks. Here are the
> ↪  actions I can do
>
> Pick up a block
> Unstack a block from on top of another block
> Put down a block
> Stack a block on top of another block
>
> I have the following restrictions on my actions:
> I can only pick up or unstack one block at a time.
> I can only pick up or unstack a block if my hand is empty.
> I can only pick up a block if the block is on the table and the block is clear. A block is clear
> ↪  if the block has no other blocks on top of it and if the block is not picked up.
> I can only unstack a block from on top of another block if the block I am unstacking was really on
> ↪  top of the other block.
> I can only unstack a block from on top of another block if the block I am unstacking is clear.
> Once I pick up or unstack a block, I am holding the block.
> I can only put down a block that I am holding.
> I can only stack a block on top of another block if I am holding the block being stacked.
> I can only stack a block on top of another block if the block onto which I am stacking the block
> ↪  is clear.
> Once I put down or stack a block, my hand becomes empty.
> Once you stack a block on top of a second block, the second block is no longer clear.
> ```

## A.6.2 Progression Proof Prompt

---

### Progression Proof Chain of Thought Prompting and GPT-4 Response

The plan correctness is defined in terms of states resulting from executing the actions in the
↪  plan. An action is executable in a state when all its preconditions hold in that state. The
↪  state resulting from the action execution consists of everything in the previous state with
↪  the addition and deletion of add and delete effects of the action. Plan correctness is defined
↪  as follows: if the first action in the plan is applicable in the initial state, i.e., its
↪  preconditions are all present there; and the second action is applicable in the state
↪  resulting from applying the first action to the initial state, this process continues until
↪  the state resulting from the application of the last action in the last but one state gives
↪  rise to the final state where all the goals are satisfied.

Here is an example problem:

[STATEMENT]
As initial conditions I have that, Block A is clear, Block B is clear, Block C is clear, the hand
↪  is empty, Block B is on top of Block D, Block A is on the table, Block D is on the table and
↪  Block C is on the table.
My goal is to have that Block D is on top of Block B.

My plan is as follows:

[PLAN]
1. Current State: Block A is clear, Block B is clear, Block C is clear, the hand is empty, Block B
↪  is on top of Block D, Block A is on the table, Block C is on the table and Block D is on the
↪  table
   Action: unstack Block B from on top of Block D
   Reason: The above action is applicable in the current state because its preconditions; Block B
   ↪  is clear, the hand is empty and Block B is on top of Block D, are satisfied in the current
   ↪  state.
   Resulting State: Block A is clear, Block C is clear, Block D is clear, the hand is currently
   ↪  holding Block B, Block A is on the table, Block C is on the table and Block D is on the
   ↪  table

2. Current State: Block A is clear, Block C is clear, Block D is clear, the hand is currently
↪  holding Block B, Block A is on the table, Block C is on the table and Block D is on the table
   Action: put down Block B
   Reason: The above action is applicable in the current state because its preconditions; the hand
   ↪  is currently holding Block B, are satisfied in the current state.
   Resulting State: Block A is clear, Block B is clear, Block C is clear, Block D is clear, the
   ↪  hand is empty, Block A is on the table, Block B is on the table, Block C is on the table
   ↪  and Block D is on the table

3. Current State: Block A is clear, Block B is clear, Block C is clear, Block D is clear, the hand
↪  is empty, Block A is on the table, Block B is on the table, Block C is on the table and Block
↪  D is on the table
   Action: pick up Block D
   Reason: The above action is applicable in the current state because its preconditions; Block D
   ↪  is clear, the hand is empty and Block D is on the table, are satisfied in the current
   ↪  state.
   Resulting State: Block A is clear, Block B is clear, Block C is clear, the hand is currently
   ↪  holding Block D, Block A is on the table, Block B is on the table and Block C is on the
   ↪  table

4. Current State: Block A is clear, Block B is clear, Block C is clear, the hand is currently
↪  holding Block D, Block A is on the table, Block B is on the table and Block C is on the table
   Action: stack Block D on top of Block B
   Reason: The above action is applicable in the current state because its preconditions; Block B
   ↪  is clear and the hand is currently holding Block D, are satisfied in the current state.
   Resulting State: Block A is clear, Block C is clear, Block D is clear, the hand is empty, Block
   ↪  D is on top of Block B, Block A is on the table, Block B is on the table and Block C is on
   ↪  the table

Final State: Block A is clear, Block C is clear, Block D is clear, the hand is empty, Block D is
↪  on top of Block B, Block A is on the table, Block B is on the table and Block C is on the
↪  table
The goal conditions are satisfied in the final state. Hence, the above plan is valid.
[PLAN END]

Here is another example:

[STATEMENT]
As initial conditions I have that, Block B is clear, Block B is on top of Block C, Block C is on
↪  top of Block A, the hand is empty, Block A is on the table, and Block D is on the table.
My goal is to have that Block C is on top of Block B and Block D is on top of Block C.

---

My plan is as follows:

[PLAN]
1. Current State: the Block B is clear, the Block D is clear, the hand is empty, the Block B is on
↪  top of the Block C, the Block C is on top of the Block A, the Block A is on the table and the
↪  Block D is on the table
   Action: unstack the Block B from on top of the Block C
   Reason: The above action is applicable in the current state because its preconditions; the
   ↪  Block B is clear, the hand is empty and the Block B is on top of the Block C, are satisfied
   ↪  in the current state.
   Resulting State: the Block C is clear, the Block D is clear, the hand is currently holding
   ↪  Block B, the Block C is on top of the Block A, the Block A is on the table and the Block D
   ↪  is on the table

2. Current State: the Block C is clear, the Block D is clear, the hand is currently holding Block
↪  B, the Block C is on top of the Block A, the Block A is on the table and the Block D is on the
↪  table
   Action: put down the Block B
   Reason: The above action is applicable in the current state because its preconditions; the hand
   ↪  is currently holding Block B, are satisfied in the current state.
   Resulting State: the Block B is clear, the Block C is clear, the Block D is clear, the hand is
   ↪  empty, the Block C is on top of the Block A, the Block A is on the table, the Block B is on
   ↪  the table and the Block D is on the table

3. Current State: the Block B is clear, the Block C is clear, the Block D is clear, the hand is
↪  empty, the Block C is on top of the Block A, the Block A is on the table, the Block B is on
↪  the table and the Block D is on the table
   Action: unstack the Block C from on top of the Block A
   Reason: The above action is applicable in the current state because its preconditions; the
   ↪  Block C is clear, the hand is empty and the Block C is on top of the Block A, are satisfied
   ↪  in the current state.
   Resulting State: the Block A is clear, the Block B is clear, the Block D is clear, the hand is
   ↪  currently holding Block C, the Block A is on the table, the Block B is on the table and the
   ↪  Block D is on the table

4. Current State: the Block A is clear, the Block B is clear, the Block D is clear, the hand is
↪  currently holding Block C, the Block A is on the table, the Block B is on the table and the
↪  Block D is on the table
   Action: stack the Block C on top of the Block B
   Reason: The above action is applicable in the current state because its preconditions; the
   ↪  Block B is clear and the hand is currently holding Block C, are satisfied in the current
   ↪  state.
   Resulting State: the Block A is clear, the Block C is clear, the Block D is clear, the hand is
   ↪  empty, the Block C is on top of the Block B, the Block A is on the table, the Block B is on
   ↪  the table and the Block D is on the table

5. Current State: the Block A is clear, the Block C is clear, the Block D is clear, the hand is
↪  empty, the Block C is on top of the Block B, the Block A is on the table, the Block B is on
↪  the table and the Block D is on the table
   Action: pick up the Block D
   Reason: The above action is applicable in the current state because its preconditions; the
   ↪  Block D is clear, the hand is empty and the Block D is on the table, are satisfied in the
   ↪  current state.
   Resulting State: the Block A is clear, the Block C is clear, the hand is currently holding
   ↪  Block D, the Block C is on top of the Block B, the Block A is on the table and the Block B
   ↪  is on the table

6. Current State: the Block A is clear, the Block C is clear, the hand is currently holding Block
↪  D, the Block C is on top of the Block B, the Block A is on the table and the Block B is on the
↪  table
   Action: stack the Block D on top of the Block C
   Reason: The above action is applicable in the current state because its preconditions; the
   ↪  Block C is clear and the hand is currently holding Block D, are satisfied in the current
   ↪  state.
   Resulting State: the Block A is clear, the Block D is clear, the hand is empty, the Block C is
   ↪  on top of the Block B, the Block D is on top of the Block C, the Block A is on the table
   ↪  and the Block B is on the table

Final State: the Block A is clear, the Block D is clear, the hand is empty, the Block C is on top
↪  of the Block B, the Block D is on top of the Block C, the Block A is on the table and the
↪  Block B is on the table
The goal conditions are satisfied in the final state. Hence, the above plan is valid.
[PLAN END]
Here is the problem to solve:

[STATEMENT]
As initial conditions I have that: Block B is clear, Block C is clear, the hand is empty, Block B
↪  is on top of Block A, Block A is on the table and Block C is on the table.

```
My goal is for the following to be true: Block A is on top of Block B and Block C is on top of
↪   Block A.

My plan is as follows:

==================GPT-4 Response====================

[PLAN]
1. Current State: Block B is clear, Block C is clear, the hand is empty, Block B is on top of
↪   Block A, Block A is on the table and Block C is on the table
   Action: unstack Block B from on top of Block A
   Reason: The above action is applicable in the current state because its preconditions; Block B
   ↪   is clear, the hand is empty and Block B is on top of Block A, are satisfied in the current
   ↪   state.
   Resulting State: Block A is clear, Block C is clear, the hand is currently holding Block B,
   ↪   Block A is on the table and Block C is on the table

2. Current State: Block A is clear, Block C is clear, the hand is currently holding Block B, Block
↪   A is on the table and Block C is on the table
   Action: put down Block B
   Reason: The above action is applicable in the current state because its preconditions; the hand
   ↪   is currently holding Block B, are satisfied in the current state.
   Resulting State: Block A is clear, Block B is clear, Block C is clear, the hand is empty, Block
   ↪   A is on the table, Block B is on the table and Block C is on the table

3. Current State: Block A is clear, Block B is clear, Block C is clear, the hand is empty, Block A
↪   is on the table, Block B is on the table and Block C is on the table
   Action: pick up Block A
   Reason: The above action is applicable in the current state because its preconditions; Block A
   ↪   is clear, the hand is empty and Block A is on the table, are satisfied in the current
   ↪   state.
   Resulting State: Block B is clear, Block C is clear, the hand is currently holding Block A,
   ↪   Block B is on the table and Block C is on the table

4. Current State: Block B is clear, Block C is clear, the hand is currently holding Block A, Block
↪   B is on the table and Block C is on the table
   Action: stack Block A on top of Block B
   Reason: The above action is applicable in the current state because its preconditions; Block B
   ↪   is clear and the hand is currently holding Block A, are satisfied in the current state.
   Resulting State: Block A is clear, Block C is clear, the hand is empty, Block A is on top of
   ↪   Block B, Block B is on the table and Block C is on the table

5. Current State: Block A is clear, Block C is clear, the hand is empty, Block A is on top of
↪   Block B, Block B is on the table and Block C is on the table
   Action: pick up Block C
   Reason: The above action is applicable in the current state because its preconditions; Block C
   ↪   is clear, the hand is empty and Block C is on the table, are satisfied in the current
   ↪   state.
   Resulting State: Block A is clear, the hand is currently holding Block C, Block A is on top of
   ↪   Block B and Block B is on the table

6. Current State: Block A is clear, the hand is currently holding Block C, Block A is on top of
↪   Block B and Block B is on the table
   Action: stack Block C on top of Block A
   Reason: The above action is applicable in the current state because its preconditions; Block A
   ↪   is clear and the hand is currently holding Block C, are satisfied in the current state.
   Resulting State: Block C is clear, the hand is empty, Block A is on top of Block B, Block C is
   ↪   on top of Block A and Block B is on the table

Final State: Block C is clear, the hand is empty, Block A is on top of Block B, Block C is on top
↪   of Block A and Block B is on the table
The goal conditions are satisfied in the final state. Hence, the above plan is valid.
[PLAN END]

==================VALID PLAN====================
```

### A.6.3 Blocksworld Universal Algorithm Prompt

**Blocksworld Universal Algorithm Chain of Thought Prompting and GPT-4 Response**

```
Here is an example problem:

[STATEMENT]
As initial conditions I have that, Block B is clear, Block C is clear, the hand is empty, Block C
↪   is on top of Block A, Block A is on the table, Block B is on the table.
```

My goal is to have that Block A is on top of Block C and Block B is on top of Block A

My plan is as follows:

[THOUGHTS]
1. I take apart all the towers, and put all the blocks on the table.
I will check every block. Any that are already on the table can be left there. Any that are on top
↪  of another block must be put on the table, but I can only unstack them if they are clear, so I
↪  will check that before trying to unstack any of them. If a block needs to be moved but isn't
↪  clear, I'll come back to it later once it is.

Block A is already on the table but is not clear.
Block B is already on the table.
Block C is on top of Block A. I can only unstack a block that is clear. C is clear according to
↪  the initial condition, so I can proceed. I unstack it. Then, I put it down on the table. This
↪  clears Block A.

All the blocks are now clear and on the table.

2. I look at the goal and stack each tower I need, starting from the bottom. I check each of the
↪  mentioned blocks in order to determine which of them are the bases of towers.

Block A needs to be on top of Block C, so it can't be the base of a tower.
Block B needs to be on top of Block A, so it can't be the base of a tower.
Block C needs Block A on top of it. I double check the goal: this is the only condition that must
↪  hold for block C. It does not need to be on top of any other block. Therefore, I can use it as
↪  the base of a tower.

In summary, only Block C can be used as a base of a tower.

I build the tower that has Block C as a base. I don't need to move Block C, and I leave it be, as
↪  I know it is already placed correctly.
The block that goes on top of Block C is Block A. I pick up Block A. I stack it on top of Block C.
The block that goes on top of Block A is Block B. I pick up Block B. I stack it on top of Block A.
No block needs to be on top of Block B. Therefore this tower is complete.

There are no other blocks which can be used as a base of a tower, and all the blocks mentioned in
↪  the goal have been placed, so I am done constructing.

3. Now, I write down my plan in the required format:

[PLAN]
unstack Block C from on top of Block A
put down Block C
pick up Block A
stack Block A on top of Block C
pick up Block B
stack Block B on top of Block A
[PLAN END]

Here is another example:

[STATEMENT]
As initial conditions I have that, Block B is clear, the hand is empty, Block A is on top of Block
↪  D, Block B is on top of Block C, Block C is on top of Block A and Block D is on the table.
My goal is to have that Block C is on top of Block D, Block A is on top of Block C and Block B is
↪  on top of Block A.

My plan is as follows:

[THOUGHTS]
1. I take apart all the towers, and put all the blocks on the table.
I will check every block. Any that are already on the table can be left there. Any that are on top
↪  of another block must be put on the table, but I can only unstack them if they are clear, so I
↪  will check that before trying to unstack any of them. If a block needs to be moved but isn't
↪  clear, I'll come back to it later once it is.

Block A is on top of Block D. I can only unstack a block that is clear. The initial condition
↪  states that Block C is on top of Block A. Therefore, I can't unstack Block A until I move
↪  Block C. I don't touch it yet, and look at Block C.
Block C is on top of Block A. I can only unstack a block that is clear. The initial condition
↪  states that Block B is on top of Block C. Therefore, I can't unstack Block C until I move
↪  Block B. I don't touch it yet, and look at Block B.
Block B is on top of Block C. I can only unstack a block that is clear. The initial condition
↪  states that Block B is clear. I unstack it. Then, I put it down on the table. This clears
↪  Block C.
Block C is on top of Block A. I can only unstack a block that is clear. Block C is cleared
↪  previously. I unstack it. Then, I put it down on the table. This clears Block A.

```
Block A is on top of Block D. I can only unstack a block that is clear. Block A is cleared
↪  previously. I unstack it. Then, I put it down on the table. This clears Block D.

All the blocks are now clear and on the table.

2. I look at the goal and stack each tower I need, starting from the bottom. I check each of the
↪  mentioned blocks in order to determine which of them are the bases of towers.

Block C needs to be on top of Block D, so it can't be the base of a tower.
Block A needs to be on top of Block C, so it can't be the base of a tower.
Block B needs to be on top of Block A, so it can't be the base of a tower.
Block D needs Block C on top of it. I double check the goal: this is the only condition that must
↪  hold for block D. It does not need to be on top of any other block. Therefore, I can use it as
↪  the base of a tower.

In summary, only Block D can be used as the base of a tower.

I build the tower that has Block D as a base. I don't need to move Block D, and I leave it be, as
↪  I know it is already placed correctly.
The block that goes on top of Block D is Block C. I pick up Block C. I stack it on top of Block D.
The block that goes on top of Block C is Block A. I pick up Block A. I stack it on top of Block C.
The block that goes on top of Block A is Block B. I pick up Block B. I stack it on top of Block A.
No block needs to be on top of Block B. Therefore this tower is complete.

There are no other blocks which can be used as a base of a tower, and all the blocks mentioned in
↪  the goal have been placed, so I am done constructing.

3. Now, I write down my plan in the required format:

[PLAN]
unstack Block B from on top of Block C
put down Block B
unstack Block C from on top of Block A
put down Block C
unstack Block A from on top of Block D
put down Block A
pick up Block C
stack Block C on top of Block D
pick up Block A
stack Block A on top of Block C
pick up Block B
stack Block B on top of Block A
[PLAN END]
Here is the problem to solve:

[STATEMENT]
As initial conditions I have that: Block B is clear, Block C is clear, the hand is empty, Block B
↪  is on top of Block A, Block A is on the table and Block C is on the table.
My goal is for the following to be true: Block A is on top of Block B and Block C is on top of
↪  Block A.

My plan is as follows:

==================GPT-4 Response====================
[THOUGHTS]
1. I take apart all the towers, and put all the blocks on the table.
I will check every block. Any that are already on the table can be left there. Any that are on top
↪  of another block must be put on the table, but I can only unstack them if they are clear, so I
↪  will check that before trying to unstack any of them. If a block needs to be moved but isn't
↪  clear, I'll come back to it later once it is.

Block A is on the table but is not clear.
Block B is on top of Block A. I can only unstack a block that is clear. B is clear according to
↪  the initial condition, so I can proceed. I unstack it. Then, I put it down on the table. This
↪  clears Block A.
Block C is already on the table.

All the blocks are now clear and on the table.

2. I look at the goal and stack each tower I need, starting from the bottom. I check each of the
↪  mentioned blocks in order to determine which of them are the bases of towers.

Block A needs to be on top of Block B, so it can't be the base of a tower.
Block B needs Block A on top of it. I double check the goal: this is the only condition that must
↪  hold for block B. It does not need to be on top of any other block. Therefore, I can use it as
↪  the base of a tower.
Block C needs to be on top of Block A, so it can't be the base of a tower.
```

```
In summary, only Block B can be used as the base of a tower.

I build the tower that has Block B as a base. I don't need to move Block B, and I leave it be, as
↪  I know it is already placed correctly.
The block that goes on top of Block B is Block A. I pick up Block A. I stack it on top of Block B.
The block that goes on top of Block A is Block C. I pick up Block C. I stack it on top of Block A.
No block needs to be on top of Block C. Therefore this tower is complete.

There are no other blocks which can be used as a base of a tower, and all the blocks mentioned in
↪  the goal have been placed, so I am done constructing.

3. Now, I write down my plan in the required format:

[PLAN]
unstack Block B from on top of Block A
put down Block B
pick up Block A
stack Block A on top of Block B
pick up Block C
stack Block C on top of Block A
[PLAN END]
==================VALID PLAN====================
```

### A.6.4 Stacking Prompt

```
Stacking Chain of Thought Prompting and GPT-4 Response

STATEMENT]
As initial conditions I have that, Block A is clear, Block B is clear, Block A is on the table,
↪  Block B is on the table and the hand is empty.
My goal is to have that Block B is on top of Block A.

My plan is as follows:

[THOUGHTS]
1. I look at the goal and stack each tower I need, starting from the bottom.
My goal only mentions blocks B and A. I check each of the mentioned blocks in order to determine
↪  which of them are the bases of towers.

Block A needs to have Block B on top of it. I double check the goal: this is the only condition
↪  that must hold for block A. It does not need to be on top of any other block. Therefore, I can
↪  use it as the base of a tower.
Block B needs to be on top of Block A, so it can't be the base of a tower.

In summary, only Block A can be used as a base of a tower.

I build the tower that has Block A as a base. I don't need to move Block A, and I leave it be, as
↪  I know it is already placed correctly.
The block that goes on top of Block A is Block B. I pick up Block B. I stack it on top of Block A.
This satisfies the part of the goal that says that Block B is on top of Block A.
No block needs to be on top of Block B. Therefore this tower is complete.

There are no other blocks which can be used as a base of a tower, and all the blocks mentioned in
↪  the goal have been placed, so I am done constructing.

2. Now, I write down my plan in the required format:

[PLAN]
pick up Block B
stack Block B on top of Block A
[PLAN END]

Here is another example:

[STATEMENT]
As initial conditions I have that, Block A is clear, Block B is clear, Block C is clear, Block A
↪  is on the table, Block B is on the table, Block C is on the table and the hand is empty.
My goal is to have that Block A is on top of Block C and Block B is on top of Block A.

My plan is as follows:

[THOUGHTS]
1. I look at the goal and stack each tower I need, starting from the bottom.
My goal only mentions blocks A, B, and C. I am free to ignore all other blocks. I check each of
↪  the mentioned blocks in order to determine which of them are the bases of towers.
```

```
Block C needs Block A on top of it. I double check the goal: this is the only condition that must
↪  hold for block C. It does not need to be on top of any other block. Therefore, I can use it as
↪  the base of a tower.
Block A needs Block B on top of it. I double check the goal: this isn't the only condition that
↪  must hold for Block A, because Block A needs to be on top of Block C. Therefore Block A can't
↪  be the base of a tower.
Block B needs to be on top of Block A, so it can't be the base of a tower.

In summary, only Block C can be used as the base of a tower.

I build the tower that has Block C as a base. I don't need to move Block C, and I leave it be, as
↪  I know it is already placed correctly.
The block that goes on top of Block C is Block A. I pick up Block A. I stack it on top of Block C.
This satisfies the part of the goal that says that Block A is on top of Block C.
The block that goes on top of Block A is Block B. I pick up Block B. I stack it on top of Block A.
This satisfies the part of the goal that says that Block B is on top of Block A.
No block needs to be on top of Block B. Therefore this tower is complete.

There are no other blocks which can be used as a base of a tower, and all the blocks mentioned in
↪  the goal have been placed, so I am done constructing.

2. Now, I write down my plan in the required format:

[PLAN]
pick up Block A
stack Block A on top of Block C
pick up Block B
stack Block B on top of Block A
[PLAN END]
Here is the problem to solve:

[STATEMENT]
As initial conditions I have that: Block A is clear, Block B is clear, Block C is clear, the hand
↪  is empty, Block A is on the table, Block B is on the table and Block C is on the table.
My goal is for the following to be true: Block A is on top of Block B and Block C is on top of
↪  Block A.

My plan is as follows:

==================GPT-4 Response====================
[THOUGHTS]
1. I look at the goal and stack each tower I need, starting from the bottom.
My goal only mentions blocks A, B, and C. I am free to ignore all other blocks. I check each of
↪  the mentioned blocks in order to determine which of them are the bases of towers.

Block B needs Block A on top of it. I double check the goal: this is the only condition that must
↪  hold for block B. It does not need to be on top of any other block. Therefore, I can use it as
↪  the base of a tower.
Block A needs Block C on top of it. I double check the goal: this isn't the only condition that
↪  must hold for Block A, because Block A needs to be on top of Block B. Therefore Block A can't
↪  be the base of a tower.
Block C needs to be on top of Block A, so it can't be the base of a tower.

In summary, only Block B can be used as the base of a tower.

I build the tower that has Block B as a base. I don't need to move Block B, and I leave it be, as
↪  I know it is already placed correctly.
The block that goes on top of Block B is Block A. I pick up Block A. I stack it on top of Block B.
This satisfies the part of the goal that says that Block A is on top of Block B.
The block that goes on top of Block A is Block C. I pick up Block C. I stack it on top of Block A.
This satisfies the part of the goal that says that Block C is on top of Block A.
No block needs to be on top of Block C. Therefore this tower is complete.

There are no other blocks which can be used as a base of a tower, and all the blocks mentioned in
↪  the goal have been placed, so I am done constructing.

2. Now, I write down my plan in the required format:

[PLAN]
pick up Block A
stack Block A on top of Block B
pick up Block C
stack Block C on top of Block A
[PLAN END]
==================VALID PLAN====================
```

### A.6.5 Lexicographic Stacking Prompt

---

**Lexicographic Stacking Chain of Thought Prompting and GPT-4 Response**

```
Here is an example problem:

[STATEMENT]
As initial conditions I have that, Block A is clear, Block B is clear, Block A is on the table,
↪   Block B is on the table and the hand is empty.
My goal is to have that Block A is on top of Block B.

My plan is as follows:

[THOUGHTS]
1. I look at the goal and stack the tower I need, starting from the bottom.
Block B is on the table and Block B is clear.
Block A is on the table and Block A is clear, so I pick it up.
Then, I stack Block A on top of Block B. The goal conditions are satisfied.

2. Now, I write down my plan in the required format:

[PLAN]
pick up Block A
stack Block A on top of Block B
[PLAN END]

Here is another example:

[STATEMENT]
As initial conditions I have that, Block A is clear, Block B is clear, Block C is clear, Block A
↪   is on the table, Block B is on the table, Block C is on the table and the hand is empty.
My goal is to have that Block A is on top of Block B and Block B is on top of Block C.

My plan is as follows:

[THOUGHTS]
1. I look at the goal and stack the tower I need, starting from the bottom.
Block C is on the table and Block C is clear.
Block B is on the table and Block B is clear, so I pick it up.
Then, I stack Block B on top of Block C. The first goal condition is satisfied.
Block A is on the table and Block A is clear, so I pick it up.
Then, I stack Block A on top of Block B. The second goal condition is satisfied.
So all goal conditions are satisfied.

2. Now, I write down my plan in the required format:

[PLAN]
pick up Block B
stack Block B on top of Block C
pick up Block A
stack Block A on top of Block B
[PLAN END]
Here is the problem to solve:

[STATEMENT]
As initial conditions I have that: Block A is clear, Block B is clear, Block C is clear, Block D
↪   is clear, the hand is empty, Block A is on the table, Block B is on the table, Block C is on
↪   the table and Block D is on the table.
My goal is for the following to be true: Block A is on top of Block B, Block B is on top of Block
↪   C and Block C is on top of Block D.

My plan is as follows:

==================GPT-4 Response====================
[THOUGHTS]
1. I look at the goal and stack the tower I need, starting from the bottom.
Block D is on the table and Block D is clear.
Block C is on the table and Block C is clear, so I pick it up.
Then, I stack Block C on top of Block D. The first goal condition is satisfied.
Block B is on the table and Block B is clear, so I pick it up.
Then, I stack Block B on top of Block C. The second goal condition is satisfied.
Block A is on the table and Block A is clear, so I pick it up.
Then, I stack Block A on top of Block B. The third goal condition is satisfied.
So all goal conditions are satisfied.

2. Now, I write down my plan in the required format:
```

```
[PLAN]
pick up Block C
stack Block C on top of Block D
pick up Block B
stack Block B on top of Block C
pick up Block A
stack Block A on top of Block B
[PLAN END]
==================VALID PLAN====================
```

## A.7 Coinflip Prompts

### Direct CoinFlip Prompt and GPT-4 Response

```
[Instructions]
Respond only with 'yes' or 'no'. Do not include anything else in your response.
Problem to solve:

[QUESTION]
A coin is heads up. Craig flips the coin. Is the coin still heads up?

[Answer]
==================GPT-4-Turbo Response====================
no
```

### CoT CoinFlip Prompt and GPT-4 Response

```
[Instructions]
After the [Answer] tag, you may respond only with 'yes' or 'no'. Do not include anything else
↪    after that tag. The [Answer] tag must precede the final answer.

The following 3 examples are provided. Please follow the formatting used in them.

Example 0:

[QUESTION]
A coin is heads up. Craig flips the coin. Is the coin still heads up?

[Thoughts]
The coin was flipped by Craig. So the coin was flipped 1 times. The coin started heads up, so
↪    after an odd number of flips, it will be tails up. So the answer is no.

[Answer]
yes

Example 1:

[QUESTION]
A coin is heads up. Major flips the coin. Maria does not flip the coin. Is the coin still heads
↪    up?

[Thoughts]
The coin was flipped by Major. So the coin was flipped 1 times. The coin started heads up, so
↪    after an odd number of flips, it will be tails up. So the answer is no.

[Answer]
yes

Example 2:

[QUESTION]
A coin is heads up. Albert does not flip the coin. Wolf flips the coin. Page flips the coin. Is
↪    the coin still heads up?

[Thoughts]
The coin was flipped by Wolf and Page. So the coin was flipped 2 times. The coin started heads up,
↪    so after an even number of flips, it will still be heads up. So the answer is yes.
```

```
[Answer]
no

Problem to solve:

[QUESTION]
A coin is heads up. Craig flips the coin. Is the coin still heads up?

[Thoughts]
==================GPT-4-Turbo Response====================
The coin was flipped by Craig. So the coin was flipped 1 time. The coin started heads up, so after
↪  an odd number of flips, it will be tails up. So the answer is no.

[Answer]
no
```

## A.8  LastLetterConcatenation Prompts

### Direct LastLetterConcatenation Prompt and GPT-4 Response

```
[Instructions]
After the [Answer] tag, you may only respond with a lowercase string of concatenated characters.
↪  Do not include anything else after that tag. The [Answer] tag must precede the final answer.
Problem to solve:

[Question]
Take the last letters of each word in "Analy Daniel" and concatenate them.

[Thoughts]
==================GPT-4-Turbo Response====================
[Answer] yy
```

### CoT LastLetterConcatenation Prompt and GPT-4 Response

```
[Instructions]
After the [Answer] tag, you may only respond with a lowercase string of concatenated characters.
↪  Do not include anything else after that tag. The [Answer] tag must precede the final answer.

The following 3 examples are provided. Please follow the formatting used in them.

Example 0:

[Question]
Take the last letters of each word in "Anthony" and concatenate them.

[Thoughts]
The last letter of "Anthony" is y. Concatenating them is "y". The answer is y.

[Answer]
y

Example 1:

[Question]
Take the last letters of each word in "Beautiful Ash" and concatenate them.

[Thoughts]
The last letter of "Beautiful" is l. The last letter of "Ash" is h. Concatenating them is "lh".
↪  The answer is lh.

[Answer]
lh

Example 2:
```

```
[Question]
Take the last letters of each word in "Matthew River Craig" and concatenate them.

[Thoughts]
The last letter of "Matthew" is w. The last letter of "River" is r. The last letter of "Craig" is
↪  g. Concatenating them is "wrg". The answer is wrg.

[Answer]
wrg

Problem to solve:

[Question]
Take the last letters of each word in "Analy Daniel" and concatenate them.

[Thoughts]
==================GPT-4-Turbo Response====================
The last letter of "Analy" is y. The last letter of "Daniel" is l. Concatenating them is "yl". The
↪  answer is yl.

[Answer]
yl
```

## LastVowelConcatenation Prompt and GPT-4 Response

```
[Instructions]
For the purposes of these problems, a vowel is any one of the letters "a","e","i","o", or "u", but
↪  NOT "y". After the [Answer] tag, you may only respond with a lowercase string of concatenated
↪  characters. Do not include anything else after that tag. The [Answer] tag must precede the
↪  final answer.

The following 3 examples are provided. Please follow the formatting used in them.

Example 0:

[Question]
Take the last vowel of each word in "Anthony" and concatenate them.

[Answer]
o

Example 1:

[Question]
Take the last vowel of each word in "Beautiful Ash" and concatenate them.

[Answer]
ua

Example 2:

[Question]
Take the last vowel of each word in "Matthew River Craig" and concatenate them.

[Answer]
eei

Problem to solve:

[Question]
Take the last vowel of each word in "Analy Daniel" and concatenate them.
```

[Answer]

==================GPT-4-Turbo Response====================
ya

## FoomLetterConcatenation Prompt and GPT-4 Response

[Instructions]
After the [Answer] tag, you may only respond with a lowercase string of concatenated characters.
↪  Do not include anything else after that tag. The [Answer] tag must precede the final answer.

The following 2 examples are provided. Please follow the formatting used in them.

Example 0:

[Question]
Take the following words and produce a new string using them: "Anthony". The first letter of your
↪  string should be the first letter of the first word, the second letter of your string should
↪  be the second letter of the second word, and so forth. If you need the nth letter of the nth
↪  word, but that word is less than n letters long, then insert a 0 character instead.

[Thoughts]
The 1st letter of "Anthony" is A. Concatenating them is "a". The answer is a.

[Answer]
a

Example 1:

[Question]
Take the following words and produce a new string using them: "Beautiful Ash". The first letter of
↪  your string should be the first letter of the first word, the second letter of your string
↪  should be the second letter of the second word, and so forth. If you need the nth letter of
↪  the nth word, but that word is less than n letters long, then insert a 0 character instead.

[Thoughts]
The 1st letter of "Beautiful" is B. The 2nd letter of "Ash" is s. Concatenating them is "bs". The
↪  answer is bs.

[Answer]
bs

Problem to solve:

[Question]
Take the following words and produce a new string using them: "Analy Daniel". The first letter of
↪  your string should be the first letter of the first word, the second letter of your string
↪  should be the second letter of the second word, and so forth. If you need the nth letter of
↪  the nth word, but that word is less than n letters long, then insert a 0 character instead.

[Thoughts]

==================GPT-4-Turbo Response====================
The 1st letter of "Analy" is A. The 2nd letter of "Daniel" is a. The 3rd letter of "Analy" is a.
↪  The 4th letter of "Daniel" is i. The 5th letter of "Analy" is y. The 6th letter of "Daniel" is
↪  e. Concatenating them is "Aaaiye". The answer is aaaiye.

[Answer]
aaaiye

## A.9 Single Digit Arithmetic Prompts

### Arithmetic Direct Prompt and GPT-4 Response

```
[Instructions]
After the [Answer] tag, you may only respond with a single number representing the final value of
↪  the calculation. Do not include anything else after that tag. The [Answer] tag must precede
↪  the final answer.
Problem to solve:

[Question]
Simplify the following expression into a single number: 1 * (7 / (5 / (1 + (9 - (5)))))

[Answer]

==================GPT-4-Turbo Response====================
[Answer]
7
```

### Arithmetic CoT Prompt and GPT-4 Response

```
[Instructions]
After each thought, provide an intermediate answer in the form of a single number, labeled by the
↪  [Intermediate Answer n] tag, where n is replaced with the number of the intermediate answer.
↪  Do not put anything other than the intermediate answer number between the intermediate answer
↪  tag and the next thought tag (e.g. [Thought n+1]). When you are done thinking and have
↪  outputted all the requisite intermediate answers, put the [Answer] tag. After the [Answer]
↪  tag, you may only respond with a single number representing the final value of the calculation.
↪  Do not include anything else after that tag. The [Answer] tag must precede the final answer.

The following 3 examples are provided. Please follow the formatting used in them.

Example 0:

[Question]
Simplify the following expression into a single number: 2 + (4)

[Thoughts]
We simplify one set of parentheses at a time, starting from the inside.
[Thought 1]
The current form of the expression is 2 + (4).
The innermost expression is 2 + (4), which simplifies to 6.
The expression cannot be simplified further, so this will also be the final answer.
[Intermediate Answer 1]
6

[Answer]
6

Example 1:

[Question]
Simplify the following expression into a single number: 6 / (7 - (5))

[Thoughts]
We simplify one set of parentheses at a time, starting from the inside.
[Thought 1]
The current form of the expression is 6 / (7 - (5)).
The innermost expression is 7 - (5), which simplifies to 2.
[Intermediate Answer 1]
2
[Thought 2]
We plug in the previous intermediate answer into the previous expression to simplify it by one
↪  step. This gives the expression 6 / (2).
The innermost expression is 6 / (2), which simplifies to 3.
The expression cannot be simplified further, so this will also be the final answer.
[Intermediate Answer 2]
3

[Answer]
```

3

Example 2:

[Question]
Simplify the following expression into a single number: 2 / (4 - (9 - (6)))

[Thoughts]
We simplify one set of parentheses at a time, starting from the inside.
[Thought 1]
The current form of the expression is 2 / (4 - (9 - (6))).
The innermost expression is 9 - (6), which simplifies to 3.
[Intermediate Answer 1]
3
[Thought 2]
We plug in the previous intermediate answer into the previous expression to simplify it by one
↪   step. This gives the expression 2 / (4 - (3)).
The innermost expression is 4 - (3), which simplifies to 1.
[Intermediate Answer 2]
1
[Thought 3]
We plug in the previous intermediate answer into the previous expression to simplify it by one
↪   step. This gives the expression 2 / (1).
The innermost expression is 2 / (1), which simplifies to 2.
The expression cannot be simplified further, so this will also be the final answer.
[Intermediate Answer 3]
2

[Answer]
2

Problem to solve:

[Question]
Simplify the following expression into a single number: 1 * (1 * (1 * (1 + (8 / (1)))))

[Thoughts]
==================GPT-4-Turbo Response====================
We simplify one set of parentheses at a time, starting from the inside.
[Thought 1]
The current form of the expression is 1 * (1 * (1 * (1 + (8 / (1))))).
The innermost expression is 8 / (1), which simplifies to 8.
[Intermediate Answer 1]
8
[Thought 2]
We plug in the previous intermediate answer into the previous expression to simplify it by one
↪   step. This gives the expression 1 * (1 * (1 * (1 + (8)))).
The innermost expression is 1 + (8), which simplifies to 9.
[Intermediate Answer 2]
9
[Thought 3]
We plug in the previous intermediate answer into the previous expression to simplify it by one
↪   step. This gives the expression 1 * (1 * (1 * (9))).
The innermost expression is 1 * (9), which simplifies to 9.
[Intermediate Answer 3]
9
[Thought 4]
We plug in the previous intermediate answer into the previous expression to simplify it by one
↪   step. This gives the expression 1 * (1 * (9)).
The innermost expression is 1 * (9), which simplifies to 9.
[Intermediate Answer 4]
9
[Thought 5]
We plug in the previous intermediate answer into the previous expression to simplify it by one
↪   step. This gives the expression 1 * (9).
The innermost expression is 1 * (9), which simplifies to 9.
The expression cannot be simplified further, so this will also be the final answer.
[Intermediate Answer 5]
9

[Answer]
9

## Arithmetic Explicitly One-Digit Direct Prompt and GPT-4 Response

```
[Instructions]
After the [Answer] tag, you may only respond with a single number representing the final value of
↪   the calculation. Do not include anything else after that tag. The [Answer] tag must precede
↪   the final answer.

The following 3 examples are provided. Please follow the formatting used in them.

Example 0:

[Question]
Simplify the following expression into a single number: 2 + (4)
Remember: the final answer and all intermediate answers will be single digit numbers.

[Answer]
6

Example 1:

[Question]
Simplify the following expression into a single number: 6 / (7 - (5))
Remember: the final answer and all intermediate answers will be single digit numbers.

[Answer]
3

Example 2:

[Question]
Simplify the following expression into a single number: 2 / (4 - (9 - (6)))
Remember: the final answer and all intermediate answers will be single digit numbers.

[Answer]
2

Problem to solve:

[Question]
Simplify the following expression into a single number: 1 * (7 / (5 / (1 + (9 - (5)))))
Remember: the final answer and all intermediate answers will be single digit numbers.

[Answer]
==================GPT-4-Turbo Response====================
7
```

