# OpenReview forum: "Chain of Thoughtlessness? An Analysis of CoT in Planning"
_NeurIPS.cc/2024/Conference — NeurIPS 2024 poster_

### Official Review · Reviewer_4aaa · 2024-06-19

**Soundness:** 2
**Presentation:** 3
**Contribution:** 3
**Rating:** 5
**Confidence:** 5

**Summary:**

This paper evaluates the effectiveness of CoT prompting on reasoning problems within the Blocksworld domain, revealing that performance gains from CoT are limited and heavily reliant on problem-specific prompts, with diminishing returns as problem complexity increases.

**Strengths:**

- The study employs a well-defined case study within the Blocksworld domain, providing a concrete context to evaluate CoT prompting.
- The study examines the generality of examples and problem complexity offers a structured and comprehensive approach to evaluating CoT's effectiveness.

**Weaknesses:**

1. The paper ignores all experimental details except prompt. The lack of description of experimental details, including the temperature and other parameters used in the data set, affects the reproducibility of the paper.
2. The paper is not very clear. The core point of the paper is that the model CoT is very sensitive to prompts and demonstrations. However, the paper spends a lot of time exaggerating the failure of CoT-style demonstration.

**Questions:**

1. I seriously doubt the test of Table 1. Why does Lexicographic Stacking Prompt only have 17 samples (*/17), while other prompts have 270 samples? This comparison is completely unfair. The model may only perform well on these 17 examples.
2. I think there is an obvious overclaim in this paper. This paper only shows that the current CoT is very sensitive in this kind of Action-related tasks. In addition, the effectiveness of CoT plans is often verified on natural language mathematical tasks. Because the expression of mathematics + natural language is more free, and theplanning logic is more general, demonstration can be used to stimulate better output of CoT style, including various types such as ICL CoT, Zero-shot CoT, Plan-and-Solve, Complex- CoT and a series of methods.
3. In fact, as shown in Figure 1, the absolute improvement of CoT has become smaller. The essential reason is that the model itself cannot solve such a complex problem. It is not that the CoT strategy is not good. In fact, compared with direct output, the relative increase in performance is actually Improved (CoT Acc/Direct Acc)

**Limitations:**

The authors have adequately addressed the limitations.

---

> ### Author Rebuttal · Authors · 2024-08-07
>
> Thank you for the thorough review.
>
> Responses to questions:
>
> 1. The lexicographic stacking problem is a special case. For a given number of blocks, there is only one problem which requires stacking them all in lexicographic order, as the syntactic stringency of the order fully determines the problem. All of our instances are between three and 20 blocks total, thus giving us exactly 17 possible lexicographic problems to test. You are exactly correct that the model may only perform well on these examples–in fact, they are constructed to explicitly require only syntactic matching. We will make this clearer in the text and description, and we will highlight this in the final table by shading the relevant squares a different color.
>
> 2. Section 6 extends our results to previously studied natural language problems. We took domains that had been examined in the original CoT paper–coinflip and last letter concatenation–and created versions where more reasoning steps are necessary. Neither of these is “action-related”. In both cases, previous work has claimed CoT learns the necessary procedure by example. In the same section, we also examine a very simplified natural language arithmetic domain where the model only needs to repeatedly simplify one digit by one digit arithmetic expressions. We discuss the results and point out that we see the same trends as before in letter concatenation and arithmetic simplification.
>
> As for the varieties of CoT listed: In-Context Learning (ICL) CoT and Zero-shot are both explicitly covered.
>
> If by Complex CoT, the reviewer is referring Fu et al’s 2023 paper Complexity-Based Prompting for Multi-step Reasoning, then, to quote from it: “We observe a clear trend on both GSM8K and MathQA: complex prompts perform on par with simple prompts on hard cases, while achieving more clear gains on cases with fewer number of reasoning steps.“ This directly matches our results. Low reasoning step number problems are very amenable to CoT improvements, but these gains disappear when generalization is attempted.
>
> 3. We assume the reviewer is referring to Figure 2. We’re not saying that CoT doesn’t improve raw performance. We’re saying that the mechanism underlying it does not seem to involve learning the procedure or algorithm demonstrated. If it did, then we’d expect to see generalizable performance improvements. Table-to-stack problems are not particularly complex: every block is on the table, and the model is tasked with stacking them in a predetermined order.
>
> The stacking prompt proceeds very similarly to an explicit n-shot plan-and-solve prompt: first the model figures out which is the bottom block, and then it figures out, step-by-step, which block goes on top of the previously placed block. Every one of these steps requires only a simple syntactic transformation that LLMs are known to excel at, but when the model is tasked with doing this for instances larger than the ones demonstrated, it fails to extend the procedure correctly.
>
> This is even clearer in the case of last letter concatenation. There, the model need only list every word and its last letter and then concatenate them together. It is able to do this perfectly for up to a few words, but afterwards, performance quickly plummets. Our main point, however, is that CoT does not work the way that it has been described in previous work. The model may do better, but it doesn’t learn and apply a new algorithm.
>
> **About experimental details:** All code and data in this paper will be made public, together with instructions for setting up and running the same tests with any API-accessible LLM. We will add the details about the exact models and temperatures we used to the paper. They are reproduced below:
>
> Temperature was set to 0 for all experiments except those already explicitly mentioned in the current text (self-consistency and some single-digit arithmetic). We used the static models when possible: GPT-4-Turbo is gpt-4-turbo-2024-04-09 and GPT-4 is  gpt-4-1106 in the OpenAI API.
> Claude-3-Opus was accessed mid-April 2024.
>
> The instances within the Blocksworld domain were generated using the PDDL generators provided by the International Planning Competitions. Within each problem class an equal number of instances were generated across the number of blocks. The intended test set for zero-shot, progression proof and universal algorithm had a total of 270 instances (15 instances per # of blocks). For the stacking prompt, the test set had a total of 261 instances as there are only 6 stack combinations for 3 block problems. Finally, there was only one instance per # of blocks for the lexicographic case.
>
> The coinflip, lastletter, and arithmetic domains are detailed in appendices A.3, A.4, and A.5. CF and LLC are both domains extended directly from Wei’s 2022 Chain of Thought paper, modified to allow for arbitrary length instances. Exact distributions are in the appendix.
>
> The multi-step arithmetic domain is a synthetic domain. We will add the random generation procedure for problems that we used to create the dataset: generate the innermost number m; uniformly rejection sample an operation # and number n, such that n#m=k is an integer from 1 and 9; if the number of reasoning steps is enough, stop, otherwise set m<-k and jump to the rejection sampling step.
>
> **The core point of our paper** is not that CoT is sensitive to prompts and demonstrations, but that CoT, contrary to previous claims, does not in-context teach LLMs general and robust algorithmic procedures. We show that CoT depends on specific prompts being narrowly constructed and  customized to the generality of the problem class and the length complexity of the instance itself. Our results provide critical evidence that counters the current claims and consensus that CoT unlocks human-like procedural reasoning abilities within LLMs from a few demonstrations, and suggest that pattern matching rather than procedure following may be a better explanation for its success.

---

> > ### Comment · Reviewer_4aaa · 2024-08-08
> >
> > Thank you for your response. Your rebuttal indeed clarified some of my misunderstandings. I have decided to improve my overall score.
> >
> > 1. **It seems that over-claiming still exists.** As you mentioned, the CoT prompt does not make the model robustly learn the correct algorithmic procedures. However, in most of your tasks, CoT indeed shows an improvement compared to Direct, which precisely proves that LLMs can learn a step-by-step thought.
> > 2. **Prompt sensitivity:** Additionally, I still believe that the contribution of this paper is limited to the fact that CoT is only sensitive to specific prompting or relies on a certain way of thought. In fact, to solve certain types of problems, demonstration or prompt strategy is inherently domain-specific. The clearer and more relevant the logical format for some problems, the faster and better the reasoning performance can be achieved. This kind of discussion seems unnecessary. For example, if I demonstrate or guide anyone through a Program-of-Thought logic, as humans, we cannot completely follow the Program-of-Thought logic to solve commonsense reasoning problems that cannot be strictly expressed.
> > 3. **Insufficient consideration:** Based on the previous point, using a specific logic to solve specific tasks is inherently a shortcut for humans, while general logic is often inefficient for uncommon tasks with special detailed settings (including so-called blocksworld, letter concatenation, and coin flipping). I believe this is beyond doubt. Therefore, I still think it is insufficient that this paper does not discuss the benchmark, like GSM8K, MATH, CommensenseQA, which is a natural language problem requiring general logic.

---

> ### Author Response · Authors · 2024-08-09
>
> Thank you for your reply.
>
> 1. Perhaps we are misunderstanding, but this response seems contradictory: robustly learning and applying a correct algorithmic procedure and learning a “step-by-step thought” are the same thing. Our claim is that, while the improvements seen do exist, they are brittle in ways that robustly learning the requisite step-by-step process wouldn’t be–raw improvement does not contradict this. The evidence necessary for our claim is the rapid deterioration of this improvement on instances that require more reasoning steps, a phenomenon we demonstrate across multiple diverse multi-step reasoning domains, from variants of Blocksworld to last letter concatenation and arithmetic expression simplification. In the camera ready version, we will make much clearer both this distinction and how we are using the evidence we gather to show the claim we make.
>
> 2. Outside of zero-shot methods (“let’s think step by step”), manual CoT construction is always domain-specific, and requires demonstrating an in-context procedure for each exemplar. We do not claim anything about CoT’s sensitivity to particular prompts. Consider the Last Letter Concatenation domain: performance is perfect for smaller instances, yet falls quickly on larger ones, despite the prompt and procedure being the same in all cases. Our contribution is to provide critical evidence that the intuitions underlying previous work–that CoT demonstrations of procedures allow LLMs to learn those procedures and apply them in-context, an anthropomorphization arising from the response format engendered by the technique and the popular name for it–miss the mark. We constrain our claims and evaluations to CoTs that are designed for and applicable to the problems they’re presented with.
>
> 3. In its current form, our paper does explicitly discuss GSM8k, CommonsenseQA, MAWPS, AsDiv, and others. We go further in depth on arithmetical reasoning benchmarks and construct a simplified natural language synthetic benchmark in section 6, under the heading “Multi-step Arithmetic on Single Digit Numbers”. In particular, we discuss a fundamental limitation of these benchmarks: they generally require only a couple of reasoning steps. Only ten percent of all GSM8k problems (a benchmark which explicitly attempted to increase the number of reasoning steps necessary) require more than five steps, and none are over eight. This sort of narrow problem distribution is insufficient to properly evaluate whether the model has learned and applied the correct procedure or is merely pattern matching to syntactically similar templates.
>
> Note also the quote we brought up in our rebuttal from the Complex CoT paper, which we will include together with a deeper discussion in our camera ready version: “We observe a clear trend on both GSM8K and MathQA: complex prompts perform on par with simple prompts on hard cases, while achieving more clear gains on cases with fewer number of reasoning steps.” Robust generalization is still not seen even with additional prompt engineering or manipulation.
>
> Furthermore, as mentioned in our paper, CommonsenseQA and similar domains do not explicitly require multi-step reasoning. In fact, the CoT exemplars given in Wei’s 2022 Chain of Thought paper for CSQA are all exactly two steps. Because CSQA questions require broad knowledge, and are very amenable to memorization, there is no way to scale the dataset to instances that necessarily require more steps–these are problems that mainly test what the model knows rather than whether the model can reason in a generalizable manner.

---

> > ### Comment · Reviewer_4aaa · 2024-08-13
> >
> > Thanks you for your detailed response. I have no additional questions.

---

### Official Review · Reviewer_g5xe · 2024-07-04

**Soundness:** 1
**Presentation:** 1
**Contribution:** 1
**Rating:** 4
**Confidence:** 4

**Summary:**

This paper evaluates the efficacy of Chain of Thought (CoT) prompting in improving the reasoning capabilities of large language models (LLMs) in planning tasks. The authors analyze CoT's performance in the Blocksworld domain, a classical planning problem, and extend their findings to other synthetic tasks. They demonstrate that CoT prompts only show meaningful improvements when the provided examples are highly specific to the problem class, and that these improvements quickly deteriorate as the complexity of the problems increases. The paper challenges the notion that CoT enables LLMs to learn general algorithmic procedures, suggesting instead that performance gains are largely due to pattern matching rather than genuine algorithmic understanding.

**Strengths:**

1. **Comprehensive Evaluation**: The paper provides a thorough analysis of CoT prompting across various problem domains, including Blocksworld, Coin Flip, Last Letter Concatenation, and multi-step arithmetic. This breadth ensures that the findings are robust and generalizable.
2. **Detailed Analysis**: The authors dissect the performance of LLMs on different levels of problem complexity, offering insights into the limitations of CoT prompting as problem size and complexity increase.
3. **Critical Perspective**: The paper critically examines the assumptions behind CoT prompting, providing evidence that contradicts the widely held belief that CoT enables LLMs to learn and apply general reasoning strategies.

**Weaknesses:**

1. **Weak Motivation**: The Chain-of-Thought paper [1] claims that "chain-of-thought prompting, where a few chain of thought demonstrations are provided as exemplars in prompting." However, the findings from this paper, specifically in lines 84 to 88, reiterate the same point, stating that CoT prompts act as exemplars and involve "pattern matching."
2. **Unrelated Title**: The paper primarily evaluates CoT prompts in a single planning task, Blocksworld, and the results indicate that CoT is still useful, as shown in Table 1. However, the title "Chain of Thoughtlessness" does not accurately reflect the content or findings.
3. **Lack of Smooth Transitions**: The content in the introduction and related work sections lacks order, making the paper structure disjointed and difficult for readers to follow. The transitions between paragraphs are abrupt and lack coherence. For example, using subheadings in the related work section could help group related content into subsections, creating a more logical flow.
4. **Section 3 Placement**: The background information in Section 3 could be integrated into a subsection of the related work, providing a smoother introduction to the planning tasks.
5. **Unclear Conclusions**: The conclusions in lines 231 to 235 are not well-supported by the results. Table 1 shows that Zero-shot CoT, Blocksworld Universal Algorithm CoT, Stacking CoT Prompt, and Lexicographic CoT Stacking Prompt all improve performance, with some showing significant improvements. However, the paper claims that the CoT approach does not enhance performance, which contradicts my understanding. Perhaps my interpretation of this table is incorrect.
6. **Lack of Examples in Main Text**: Lines 196 to 220 should include specific examples directly in the main text, as they are central to the experiments. Currently, these examples are relegated to the appendix, which diminishes their impact and increasing the understanding of remaining sections.
7. **Insufficient Experimental Explanation**: The experimental section lacks clarity and organization. Each experiment should be clearly described, explaining what each aims to demonstrate, such as Experiment A for generality and Experiment B for complexity.
8. **Missing Statistics about Test Set**: There are no statistics provided about the Blocksworld test set, leaving the question distribution unclear. This is essential evidence for evaluating the generality of CoT prompts.
9. **Unaddressed Human Labor Cost**: The paper frequently mentions the drawback of CoT prompts requiring human labor but does not provide any comparative analysis or solutions. This point does not directly support the paper's argument and feels extraneous.
10. **Writing Typos**:
    - Line 72: PDDL is introduced as an acronym without explanation.
    - Line 239 likely refers to Figure 2, which should be corrected.

> Reference:
> [1] Wei, J., Wang, X., Schuurmans, D., Bosma, M., Xia, F., Chi, E., ... & Zhou, D. (2022). Chain-of-thought prompting elicits reasoning in large language models. Advances in neural information processing systems, 35, 24824-24837.

**Questions:**

N/A

---

> ### Author Rebuttal · Authors · 2024-08-07
>
> Respectfully, we found this review highly inconsistent and in places incoherent. We wonder if there was some transmission/saving error when the reviewer posted it. Nevertheless, let us respond to the review as it was posted. We hope our clarifications persuade you to rethink your overall evaluation of the paper.
>
> Lines 84-88 are a summary of our results: “Overall, this case study calls into question assumptions about the generalizable effectiveness of chain of thought, and suggests that LLMs do not learn new, general algorithms in context, but instead rely on some form of pattern matching to achieve prompt-design-specific performance increases.”
>
> The reviewer’s quote is from Wei’s 2022 paper introducing Chain of Thought. The full context from the paper:
> > We explore how generating a chain of thought—a series of intermediate reasoning steps—significantly improves the ability of large language models to perform complex reasoning. In particular, we show how such reasoning abilities emerge naturally in sufficiently large language models via a simple method called chain-of-thought prompting, where a few chain of thought demonstrations are provided as exemplars in prompting.
>
> Our paper directly challenges the generality of this claim. While CoT does improve performance in terms of raw accuracy in many domains, our results show that it does not do so in a way that generalizes across problem complexities, despite including explicit demonstrations of algorithms that do generalize along these dimensions. Thus, we position our paper as a counterpoint to Wei’s “lower bound” approach–our results give evidence about the upper bound of CoT efficacy as part of a critical examination of the assumption that CoT unlocks human-like procedural reasoning from a few examples.
>
> There may be some confusion about the meaning of the title. The full title is “Chain of Thoughtlessness? An Analysis of CoT in Planning”. The word “thoughtlessness” does not refer to the opposite of thought, but to the opposite of thoughtfulness. What we are questioning with this title is whether “chains of thought” generated by LLMs are produced in a careless way–one which does not seem to involve generalizable procedure learning but instead looks to consist of slapdash pattern matching. We would also argue that the word “thought” in the phrase “chain of thought” is already very loaded and anthropomorphized, and so is fair to criticize.
>
> Furthermore, we examine three domains other than blocksworld: coinflip, last letter concatenation, and an arithmetic benchmark. The first two are direct extensions of previous work that made explicit claims about the breakthrough efficacy of CoT on these domains. We confirm that CoT prompting does generalize fairly far in the coinflip domain, but find that on last letter concatenation and arithmetic expression simplification, performance drops substantially when the domain is extended, showcasing that the model did not correctly replicate the procedure demonstrated in the prompt.
>
> We wish to reiterate that the central claim of our paper is not that CoT does not lead to any performance improvement, but, to quote the reviewer’s strengths section, to offer evidence against “the widely held belief that CoT enables LLMs to learn and apply general reasoning strategies.” We will clarify this point further in the final copy, and we will change the phrase “does not meaningfully enhance performance” to “does not robustly generalize.”
>
> We’d also like to point out that, in table 1, Zero-shot CoT improves or worsens (depending on the model) performance by around one percentage point, making no meaningful difference, and that the Blocksworld Universal algorithm is already a fairly narrow prompt which can only perform well on a loose relaxation of the domain. Note that the improvements become more significant the more granular and problem-specific the prompts become–which is a point we make in the paper: to effectively utilize CoT, the user needs to greatly restrict the problem or write different CoTs for many subproblems. We will add this analysis to the appendix.
>
> Blocksworld instances were generated using PDDL generators provided by the International Planning Competitions. The intended test set for zero-shot, progression proof and universal algorithm had a total of 270 instances (15 instances per # of blocks between 3 and 20). For the stacking prompt, the test set had a total of 261 instances as there are only 6 stack combinations for 3 block problems. Finally, there was only one instance per # of blocks for the lexicographic case. We will add these details to the appendix.
>
> On formatting:
>
> - Almost every paragraph begins with a linking sentence introducing the new topic. We will also improve the prose: lines 40-47 will be rearranged to improve flow, lines 48-53 will be mostly removed. We can add headings to the paragraphs in the related work as follows: What is CoT?, Enhancements to CoT, Problems with CoT, Generalizability of CoT, Current Opinions. We will also rework the final paragraph of the related work section to increase clarity and readability.
> - Section 3 will be a subsection of related work.
> - Due to the amount of detail provided in each prompt, it is infeasible to provide examples in the main body. We compromise by putting the prompts in the appendix and creating figure 1, which illustrates each of the three levels of generality in the blocksworld domain. We will make it clearer in the text which sections corresponds to which problem.
> - We do provide a definition of PDDL in the background section on line 146, but we will add a parenthetical to the introduction to clarify.
> - We have corrected the figure references.

---

> > ### Comment · Reviewer_g5xe · 2024-08-08
> >
> > Thank you for the authors' response.
> > First, I apologize for the unstructured review. My main concern is that your paper seems to overclaim, and the experiments are not convincing, which has also been mentioned by other reviewers. I will wait for the responses from the other reviewers before deciding whether to raise the score.

---

> ### Author Response · Authors · 2024-08-09
>
> Thank you for your reply.
>
> We reiterate that our global response as well as the response to you clearly points out that we are only claiming that CoT doesn’t robustly learn the algorithmic procedure demonstrated in prompt, as shown by the performance deteriorating as the number of reasoning steps increases (despite the fact that the given procedure does solve the problem). This is not contradicted by raw improvements on problems similar to the given CoT exemplars. As we pointed out in our individual response to you, your original review seems to have misread this.
>
> We would also like to draw your attention again to the fact that in addition to planning tasks, we have done experiments on extensions of standard CoT benchmarks–including last letter concatenation (Section 6)--and those results too are in agreement with our claim about Cot not leading to general procedure learning.
>
> Finally, we note–as you yourself can readily see–that all the other reviewers have had significantly more positive assessment of the paper.
>
> We look forward to your reconsideration of your assessment.

---

### Official Review · Reviewer_zEQi · 2024-07-12

**Soundness:** 4
**Presentation:** 4
**Contribution:** 3
**Rating:** 8
**Confidence:** 4

**Summary:**

The paper conducts a systematic study of claims that chain-of-thought (CoT) prompting unlocks reasoning abilities in LLMs. In particular, the paper evaluates the ability of LLMs to a) learn a simple algorithm from demonstrations annotated with reasoning steps ("thoughts") provided as part of the input prompt, and b) to generalize to harder instances in the same problem class. CoT prompt variants are evaluated on a carefully constructed set of simple planning problems (e.g., Blocksworld) and simplified variants. The experimental evaluation demonstrates that CoT works better when the prompt includes reasoning demonstrations on examples "similar" to the query (test) problem and when the test problem class is sufficiently easy (small, specific). The paper makes the case that CoT enables a form of syntactic pattern matching rather than algorithmic reasoning. Overall, the paper provides deep empirical insight into CoT prompting, clearly demonstrating its limited ability to generalize to larger tasks requiring multi-step reasoning when given smaller sized demonstrations, a task easily handled by sound planning algorithms.

**Strengths:**

+ The paper tackles an important topic of large interest to the community. The extent of LLMs abilities aren't fully understood, especially on challenging tasks (reasoning, planning) and it is important to carefully assess claims of new abilities on these tasks.

+ The paper performs a rigorous empirical evaluation using the classical and easily-understood domain of Blocksworld as well as other domains. By evaluating a variety of general-to-specific prompts across problem distributions, the paper is able to thoroughly assess claims of reasoning abilities.

+ The result of this careful evaluation is strong evidence of the limited ability of CoT to induce algorithmic reasoning in LLMs using only annotated demonstrations. Rather, some form of syntactic pattern matching seems to be occurring.

+ The paper is extremely well written and easy to understand. The appendixes are richly detailed.

**Weaknesses:**

- I didn't spot any major weaknesses in this paper. Some minor nitpicks are in the questions.

**Questions:**

- This may be out of scope but I'd be curious to understand how much variance there is in these results wrt to the prompt inputs. Specifically, how robust are the results wrt (semantically) small changes to the system prompt, examples, thoughts, formatting, etc.?

- (Line 200) I'm probably missing something but where exactly is the meta prompt explaining plan correctness in A.6.2?

- Some references seem to be incorrect or missing.
  - (Line 239) Should it be "Figure 2" (instead of "Figure 3")?
  - (Line 253) Should it be "Table 2" (instead of "Table 3")?
  - Should Figure 3 and Table 3 be referenced somewhere in Sec 6.1?

**Limitations:**

Yes

---

> ### Author Rebuttal · Authors · 2024-08-07
>
> Thank you for the thoughtful review.
>
> The progression proof CoT prompt does include a meta-prompt, but we failed to include it in the original draft. Here it is:
> > The plan correctness is defined in terms of states resulting from executing the actions in the plan. An action is executable in a state when all its preconditions hold in that state. The state resulting from the action execution consists of everything in the previous state with the addition and deletion of add and delete effects of the action. Plan correctness is defined as follows: if the first action in the plan is applicable in the initial state, i.e., its preconditions are all present there; and the second action is applicable in the state resulting from applying the first action to the initial state, this process continues until the state resulting from the application of the last action in the last but one state gives rise to the final state where all the goals are satisfied.
> We will add this to the appendix.
>
> We’ve also fixed all the missing and incorrect references, including the ones you mentioned. Thank you for catching these!
>
> As to your question, we did also test multiple prompts across our experiments, to ensure that minor details of prompt selection did not impact our results, and chose the best performing ones for fairness. In the Blocksworld domains, we tried four varieties of the universal prompt, ranging from more to less explicit algorithm demonstrations. We compared a CoT that required finding the base block first and ones that instead serialized the goal conjuncts, eventually settling on the prompt that gave the best performance. We also compared varieties of lexicographic prompt while looking for a clearcut example where CoT guarantees length generalization over n-shot prompting, and we checked reverse lexicographic problems, as well as problems where the given query had the opposite order of the prompts given. We found some variation across all of these, but nothing that contradicted the general trends we observed in our reported findings. All data we gathered, whether featured in the main body of the paper or not, will be publicly released on github, and we will add a discussion of these prompt variations to the appendix.
>
> We also considered prompt variations in the non-planning domains:
>
> In the coin flip and last letter concatenation domains, we generated the same kinds of prompts that Wei’s 2022 Chain of Thought paper used. However, as we needed more varied names, we drew ours from a different source (the US Social Security administration, as discussed in appendices A.3 and A.4), and partitioned these into 1, 2, and 3-length names. We measure the length as the number of tokens the GPT-4-Turbo tokenizer (CL100K_Base) requires to encode them. We found no significant differences in performance between them, and included equal mixtures of all three kinds of prompts in our reported experiments.
>
> We also varied the number of examples–from 1 to 3–and the correctness of the examples: either all correct examples or all incorrect. While incorrect examples do sometimes show a small decrease in performance, it is almost the same, and the overall trend is unaffected (see also Invalid Logic, Equivalent Gains, Schaeffer et. al. 2023 for a more thorough analysis of wrong CoTs).
>
> In the letter concatenation task, we also tried a prompt where, instead of saying “let’s think step by step” we merely added a “[Thought]” tag to the end of the direct zero-shot prompt. This retained some of the improvement of normal zero-shot CoT, but didn’t do quite as well.
> In the arithmetic task, we varied the requirements of the CoT from free-form thoughts to requiring intermediate answers to be tagged to requiring intermediate answers and computations to be tagged. We also tried explicitly including an instruction that every intermediate computation will be a single digit integer. Performance was roughly the same across all, and the overall downward trend was unaffected by these modifications.
>
> We will add these additional experiments and associated charts together with this discussion to the appendix.

---

> > ### Comment · Reviewer_zEQi · 2024-08-13
> > **Re. author response**
> >
> > I thank the authors for their detailed response to all reviewers. After reading the other reviews and the comments, I'm now more positive about the paper. In my opinion, the authors have been clear about their central claims both in the paper and comments ("we are only claiming that CoT doesn’t robustly learn the algorithmic procedure demonstrated in prompt, as shown by the performance deteriorating as the number of reasoning steps increases"), and proceed to demonstrate its empirical validity rigorously. I think this paper adds valuable empirical insight into the limitations of current LLMs, which is sufficient for me to recommend acceptance. Beyond that, I think this paper would be a valuable addition to the growing body of work in LLM-based planning.

---

### Official Review · Reviewer_x4qq · 2024-07-17

**Soundness:** 3
**Presentation:** 4
**Contribution:** 3
**Rating:** 7
**Confidence:** 4

**Summary:**

The paper aims to show that Chain of Thought style prompting does not result in generalisation of reasoning, instead relying on pattern matching to improve performance. It argues that if CoT results in language models learning algorithms in context, then prompts describing general procedures should result in similar performance gains to prompts describing task specific examples. Experiments are performed on planning problems in the Blocksworld domain using prompts at different levels of generality. The results show smaller performance gains from general prompts compared to large gains for more specific prompts. Through further experiments on planning problems in other domains, the paper shows that CoT does not generalise to more complex problems than those presented as examples. The paper concludes that improvements from CoT are due to pattern matching rather than in-context learning of algorithms.

**Strengths:**

1. Novel and interesting evaluation of how the performance gains from CoT can vary with the level of generality of CoT described in the prompt.
2. Comprehensive evaluation of how CoT fails to generalise to problems of higher complexity across a variety of tasks.
3. The blocksworld experiments as well as other synthetic benchmarks are a useful contribution for evaluating reasoning in a scalable manner.

**Weaknesses:**

While the results presented do indicate that more general demonstrations of CoT do not perform as well as task specific ones, they also do show improvement over standard prompting. For example in Table 1, GPT4 jumps from 7% to 28% using the Blockworld Universal Algorithm. A similar trend is present in Table 2 and Table 3. It is unclear to what extent this is from pattern matching or from algorithmic generalisation. The paper may be at risk of overstating their claim that CoT is not inducing any reasoning. Discussion and further analysis of this would be helpful.

A concern about the "Progression Proof CoT" prompt: The prompt includes demonstrations of successful plans with details of the actions and states. However, it does not include any reasoning / algorithmic description of how the plan was obtained from the problem. To be considered a legitimate Chain of "Thought" prompt, should not this prompt include description of a general procedure to derive the plans? Otherwise, it is difficult to see of the LLM can be expected to generalise reasoning from a description of the final answer without intermediate reasoning "thoughts".

Overall, a novel and interesting contribution but there are some concerns about evaluation and the conclusions drawn from them.

**Questions:**

1. In 5.2, the paper says "only the most specific and least applicable prompts retain anywhere near this performance improvement". Is this referring to the "Blocksworld Universal Algorithm" or the "Stacking Prompt"? In the case of the former, is this also not a fairly general prompt? More broadly my concern is that Table 2 shows that the Universal Algorithm Prompt achieves similar performance to the Stacking prompt, showing that a more general CoT can perform as well as a more specific one, which is contrary to the central claim of the paper.

2. The performance of GPT4 on Lexicographic Stacking seems surprisingly low with the n-shot prompt (considering the CoT prompt achieves 94%). Was this due to the particular examples provided in the n-shot prompt?

3. Table 3 does not seem to be referenced in the text?

4. Does 5.1 contain an incorrect reference to Figure 3?

5. Does 5.2 contain an incorrect reference to Table 3?

6. Can the n-shot prompts be included in the appendix?

**Limitations:**

I could not find any discussion of the limitations of the work. One limitation is that the work does not provide methodologies for reasoning with LLMs beyond that go beyond the limitations described.

---

> ### Author Rebuttal · Authors · 2024-08-07
>
> Thank you for the thorough review.
>
> Analyzing the data using only the tables is somewhat misleading. Figures 2 and 3, as well as the appendix figure A.1.1 show more clearly that the bulk of the improvement for all prompts is on the few-block problems, whereas if the procedure shown in these CoTs were followed, we would expect this improvement to be much more robust on examples of larger length than those shown in the examples.
>
> **To the point about the progression proof prompt:** “Chain of Thought” is a loosely defined term in the literature. Intermediate “thoughts” vary from piece-by-piece rationales to compositional reasoning steps. The progression proof CoT we use is taken from previous literature that studied LLM performance on this domain ([1] call it a “state-tracking CoT prompt”). At each step, this prompt provides not just reasons for the given action, but also the current state from which to determine which action to take next. In their 2023 survey of CoT techniques, Zhang et. al. specify that “the intermediate processes of CoT reasoning [...] can encompass solutions, intermediate reasoning steps, or any relevant external knowledge pertaining to a question.” The progression proof provides partial solutions and their effects, thus making it a valid CoT instantiation.
>
> In fact, the procedure provided, if followed, should ensure that the output plan is valid. Yes, this prompt doesn’t provide an algorithm that precisely specifies every part of the reasoning process, but neither does any CoT–all CoTs assume that the model can handle some parts automatically. For example, in grade school arithmetic tasks, the selection of relevant numbers and the exact sequencing of the steps needed to solve the problem (which can be the hardest part of the problem, something which is exploited by teachers who introduce unnecessary additional information) is not specified in the chain of thought prompts provided in the original Chain of Thought paper (Wei et. al. 2022). Note also that, while the progression proof guarantees the executability of the output plan, we have data showing that this guarantee is not respected by the LLMs we test, thus showing they did not execute the given procedure:
>
> If they had, they would have generated a high or perfect percentage of plans that were executable. GPT-4-Turbo generated 9.36% executable plans, Claude-3-Opus was 59.63%, and GPT-4 was 38.15%. We will include these figures in the appendix as additional analysis of the LLMs’ inability to execute the procedures demonstrated in the CoT.
>
> **The central claim of our paper** is that, because CoT prompts do not lead to the model robustly learning the correct algorithmic procedure, effective CoT prompts require both strong specificity to the problem class and specificity to the length complexity of the problem itself. Our data generally supports the problem class specificity hypothesis, as zero shot CoT does worse than the progression proof, the progression proof CoT generally does worse than the universal algorithm, the three-part universal algorithm does worse on two out of three models than the two-part stacking prompt (which is itself a version of the universal algorithm that assumes all blocks are already on the table, but is otherwise identical to it–hence how close their performance is), and the stacking prompt does worse than the lexicographic prompt, which does the best. The other part of our central claim–that CoT demonstrations do not induce length generalization–is supported not just across all but the most trivial (the lexicographic case) of our blocksworld domains, but is also shown very clearly by our analysis of the last letter concatenation and arithmetic expression simplification tasks (see Figure 3).
>
> **On lexicographic stacking performance:** In all of our problems, both the n-shot and the CoT prompts use the same examples. The lexicographic n-shot prompts tend to fail because the model neglects exactly this part of the rules: “I can only pick up a block if the block is on the table and the block is clear. A block is clear if the block has no other blocks on top of it and if the block is not picked up.” Instead the model will stack A on B, then pick up B (which is illegal based on the explicit rule cited because B is not clear) and stack it on C, and so forth. GPT-4 fails all lexicographic problems which are larger than the examples given in this exact same manner. The CoT prompt specifies a two-part procedure: first determine which block is the base of the tower, then stack all the blocks in the correct order on that base. This is sufficient to correct the issue in most, but not all, cases, most likely by aligning the most probable completion with the correct semantics in this particular case.
>
> **Typos and formatting issues:** Thank you for pointing these out! Missing references have been added and incorrect ones have been fixed.
>
> We will add the n-shot prompts to the appendix, as well as releasing the entire codebase and data publicly on github.
>
> [1] Valmeekam K, Marquez M, Sreedharan S, Kambhampati S. On the planning abilities of large language models-a critical investigation. Advances in Neural Information Processing Systems. 2023 Dec 15;36:75993-6005.

---

> > ### Comment · Reviewer_x4qq · 2024-08-14
> >
> > Thanks to the authors for their response.
> >
> > After considering the clarifications in the rebuttal, I better understand the paper's claim that CoT does not robustly generalize underlying algorithms from prompts. I believe this has been demonstrated through sufficient emperical analysis and is a valuable contribution. I have decided to increase my score given these considerations.

---

### Author Rebuttal · Authors · 2024-08-07

In addition to the reviewer-specific rebuttals, we provide this global response to a couple of common ways the reviewers misunderstood the claims/contributions of our paper.

**Central claim of our paper:**
The central claim of our paper is not that CoT can’t improve raw accuracy on static benchmarks. We demonstrate that CoT depends on specific prompts being narrowly constructed and customized to the generality of the problem class and the length complexity of the instance itself. Our experiments show that while CoT leads to performance improvements for the narrow problem classes it is tailored to, the model fails to robustly learn the demonstrated procedure/algorithm that would allow it to generalize to other problems which are also solved by that procedure. Our results provide critical counterevidence to the current consensus that CoT unlocks human-like procedural reasoning abilities within LLMs from a few demonstrations [1]. This suggests that CoT’s success is guided more by pattern matching than procedure following.

**Going beyond planning:**
While our initial interest was understanding the limitations of CoT in planning, we believe that the limitations we unearth are fundamental to CoT and also apply to other problems. Specifically, we extend our experiments to extensions of domains that have been studied by the original CoT paper [1] (Coinflip and Last Letter Concatenation), creating instances within those domains where more reasoning steps are required. As grade school arithmetic is a very commonly studied domain in the CoT literature, we created a much-simplified synthetic natural language arithmetic domain.This domain requires simplifying parenthesized expressions by repeatedly applying one of the four basic arithmetical operations on one digit numbers. All intermediate answers stay one digit, so that the only math required consists of composing operations we know every LLM can evaluate perfectly. Results on Last Letter Concatenation and One-digit arithmetic are consistent with those of Blocksworld, showcasing a performance collapse that suggests CoT prompting has failed to teach the model the necessary underlying procedure to generalize properly.

**Tables vs. Plots:**
While we gave the results of our experiments both in the form of tables and plots, we note that the plots provide a lot more insight into the performance of CoT. One trend that the plots show very clearly is how the effectiveness of CoT degrades as the problem size increases past that of exemplars in the CoT prompt. This can be seen across both planning and non-planning instances (such as last letter concatenation), and provides strong evidence that CoT does not generally induce models to learn and apply the procedure demonstrated in the prompt.

[1] Wei J, Wang X, Schuurmans D, Bosma M, Xia F, Chi E, Le QV, Zhou D. Chain-of-thought prompting elicits reasoning in large language models. Advances in neural information processing systems. 2022 Dec 6;35:24824-37.

---

### Decision · Program_Chairs · 2024-09-25

**Decision:**

Accept (poster)

**Comment:**

The paper studies whether Chain-of-Thoughts (Cot) enables generalization to larger problems involving a number of reasoning steps larger than most datasets used for reasoning.

There is a large disagreement among the reviewers’ ratings, from 4 to 8. The strongest critique was about the conclusions. Some reviewers remarked that CoT was actually helping in comparison with the direct approach. The rebuttal clarified the goal of the submission: to challenge that CoT unlocks reasoning abilities. This is done by testing on domains requiring more steps. The cases where the improvement was more significant were in more straightforward tasks like lexicographic stacking.

In my opinion, the authors clarified the scope of their contribution and showed what changes would be made so it’s more clear in a future version.